# PROGRESSIVE FOURIER NEURAL REPRESENTATION FOR SEQUENTIAL VIDEO COMPILATION

**Haeyong Kang, Jaehong Yoon, DaHyun Kim, Sung Ju Hwang, and Chang D. Yoo**[*]
Korea Advanced Institute of Science and Technology (KAIST)
{haeyong.kang, jaehong.yoon, dahyun.kim, sjhwang82, cd_yoo}@kaist.ac.kr

## ABSTRACT

Neural Implicit Representation (NIR) has recently gained significant attention due to its remarkable ability to encode complex and high-dimensional data into representation space and easily reconstruct it through a trainable mapping function. However, NIR methods assume a one-to-one mapping between the target data and representation models regardless of data relevancy or similarity. This results in poor generalization over multiple complex data and limits their efficiency and scalability. Motivated by continual learning, this work investigates how to accumulate and transfer neural implicit representations for multiple complex video data over sequential encoding sessions. To overcome the limitation of NIR, we propose a novel method, *Progressive Fourier Neural Representation (PFNR)*, that aims to find an adaptive and compact sub-module in Fourier space to encode videos in each training session. This sparsified neural encoding allows the neural network to hold free weights, enabling an improved adaptation for future videos. In addition, when learning a representation for a new video, PFNR transfers the representation of previous videos with frozen weights. This design allows the model to continuously accumulate high-quality neural representations for multiple videos while ensuring lossless decoding that perfectly preserves the learned representations for previous videos. We validate our PFNR method on the UVG8/17 and DAVIS50 video sequence benchmarks and achieve impressive performance gains over strong continual learning baselines. The PFNR code is available at https://github.com/ihaeyong/PFNR.git.

## 1 INTRODUCTION

Neural Implicit Representation (NIR) (Chen et al., 2021a; Li et al., 2022; Chen et al., 2023; Mehta et al., 2021) is a research field that aims to represent complex data, such as videos or 3D objects, as continuous functions learned by neural networks. Instead of explicitly describing data points, NIR models compress high-dimensional data from a low-dimensional embedding space. This process enables efficient data storage, compression, and synthesis. However, there's a challenge: when compressing multiple pieces of data, each high-dimensional data needs to be encoded in the neural network, increasing linear memory requirements. To address this, Neural Video Representation proposes a solution, as explored in studies by Chen et al. (2021a; 2022b). This approach combines different videos into a single video format and then reduces the model size through techniques like weight pruning and quantization post-training. While this method is effective for current data compression, it has a significant limitation: it restricts the model's ability to adapt to new videos as they are added. To overcome this limitation, inspired by incremental knowledge transfer and expansion in continual learning, we investigate a practical implicit representation learning scenario with video data, which aims to accumulate neural implicit representations for multiple videos into a single model under the condition that videos are incoming sequentially.

Continual Learning (CL) (Thrun, 1995; Rusu et al., 2016; Zenke et al., 2017; Hassabis et al., 2017) is a learning paradigm where a model learns over multiple sequential sessions. It seeks to mimic human cognition, characterized by the ability to learn new concepts incrementally throughout a lifetime without the degeneration of previously acquired functionality. Yet, incremental training of

---

[*]Corresponding author

NIR is a challenging problem since the model detrimentally loses the learned implicit representations of past session videos while encoding newly arrived ones, a phenomenon known as *catastrophic forgetting* (McCloskey & Cohen, 1989). This issue particularly matters as neural representation methods for videos encode and reconstruct the target data stream conditioned to its frame indices. Then, the model more easily ruins its generation ability while learning to continuously encode new videos due to the distributional disparities in *holistic videos* and their *individual frames*. Furthermore, the *compression phase* of neural representation makes it wayward to transfer the model to future tasks. Various approaches have been proposed to address catastrophic forgetting during continual learning, which are often conventionally classified as follows: (1) *Regularization-based methods* (Kirkpatrick et al., 2017; Chaudhry et al., 2020; Jung et al., 2020; Titsias et al., 2020; Mirzadeh et al., 2021) aim to keep the learned information of past sessions during continual training aided by sophisticatedly designed regularization terms, (2) *Architecture-based methods* (Yoon et al., 2018; Mallya et al., 2018; Serrà et al., 2018; Wortsman et al., 2020; Kang et al., 2022a;b) propose to minimize the inter-task interference via newly designed architectural components, and (3) *Rehearsal-based methods* (Rebuffi et al., 2017; Chaudhry et al., 2019b; Saha et al., 2021; Yoon et al., 2022; Sarfraz et al., 2023) involves replaying real or synthesized data from previous sessions. However, these methods are less suitable for video data in CL due to the substantial memory and computational costs required to store and revisit high-dimensional samples. While conventional architecture-based methods offer solutions to prevent forgetting, they are unsuited for sequential complex video processing as they reuse a few or all adaptive parameters without finely discretized operations.

To enhance neural representation incrementally on complex sequential videos, we propose a novel sequential video compilation method, coined **P**rogressive **F**ourier **N**eural **R**epresentation (**PFNR**) to identify and utilize Lottery tickets (i.e., the weights of complicated oscillatory signals) in frequency space. To achieve this, we define **F**ourier **S**ubnetwork **O**perator (**FSO**), which breaks down a neural implicit representation into its sine and cosine components (real and imaginary parts) and then selectively, identifies the most effective *Lottery tickets* for representing complex periodic signals. In practice, given a backbone and FSO architecture, our method continuously learns to identify input-adaptive subnetwork modules and encode each new video into the corresponding module during sequential training sessions. Our approach draws inspiration from the *Lottery Ticket Hypothesis (LTH)* (Frankle & Carbin, 2019), which suggests that sparse subnetworks can maintain the performance of a dense network and from the *Fourier Neural Operator* concepts developed in studies by Li et al. (2020a;b); Kovachki et al. (2021); Tran et al. (2021). A challenge in this domain is the inefficiency of continually searching for optimal subnetworks in Fourier space. This process typically requires iterative training steps with repeated pruning and retraining for each new task. To address this, PFNR introduces a parametric score function. This function learns to produce binary masks for the real and imaginary components, enabling the identification of adaptive substructures for video encoding in each training session by selecting the top-percentage weights based on their ranking scores. This allows PFNR to discover the optimal subnetwork during training, through joint training of weights and structure, thus avoiding the laborious processes of iterative retraining, pruning, and rewinding inherent in *LTH*. Crucially, PFNR permits overlapping subnetworks with those from previous sessions during training. This overlap allows the transfer of learned representations from earlier videos when relevant while keeping the weights for these earlier sessions fixed. As a result, our model can continuously expand its representation space across successive video sessions, ensuring that it maintains the encoding and generation quality of previous videos without any degradation (i.e., remaining forgetting-free). This is achieved without needing a replay buffer to store multiple high-dimensional frames, a significant advancement in the field.

Our contributions can be summarized as follows:

- We suggest a practical learning scenario for neural implicit representation where the model encodes multiple videos continually in successive training sessions. Earlier NIR methods suffered from poor transferability to new videos due to the distributional shift of holistic video and frames.

- We propose a cutting-edge method referred to as the Progressive Fourier Neural Representation for a complex sequential video compilation. The proposed method continuously learns a compact subnetwork for each video session given a supernet backbone while preserving the generative quality of previous videos flawlessly in Fourier space.

- We demonstrate the effectiveness of our method on multiple sequential video sessions by achieving superior performance over conventional baselines in average PSNR and MS-SSIM without

any quantitative or qualitative degeneration in reconstructing previously encoded videos during sequential video compilation.

## 2 RELATED WORKS

**Neural Implicit Representation (NIR).** Neural Implicit Representations (NIR) (Mehta et al., 2021) are neural network architectures for parameterizing continuous, differentiable signals. Based on coordinate information, they provide a way to represent complex, high-dimensional data with a small set of learnable parameters that can be used for various tasks such as image reconstruction (Sitzmann et al., 2020; Tancik et al., 2020), shape regression (Chen & Zhang, 2019; Park et al., 2019), and 3D view synthesis (Mildenhall et al., 2021; Schwarz et al., 2020). Instead of using coordinate-based methods, NeRV (Chen et al., 2021a) proposes an image-wise implicit representation that takes frame indices as inputs, enabling fast and accurate video compression. NeRV has inspired further improvements in video regression by CNeRV (Chen et al., 2022b), DNeRV (He et al., 2023), E-NeRV (Li et al., 2022), and NIRVANA (Maiya et al., 2022), and HNeRV (Chen et al., 2023). A few recent works have explored video continual learning (VCL) scenarios for the NIR. To tackle non-physical environments, Continual Predictive Learning (CPL) (Chen et al., 2022a) learns a mixture world model via predictive experience replay and performs test-time adaptation using non-parametric task inference. PIVOT (Villa et al., 2022) leverages the past knowledge present in pre-trained models from the image domain to reduce the number of trainable parameters and mitigate forgetting. CPL needs memory to replay, while PIVOT needs pre-training and fine-tuning steps. In contrast, along with the conventional progressive training techniques (Rusu et al., 2016; Cho et al., 2022), we introduce a novel neural video representation referred to as *"Progressive Fourier Neural Representation (PFNR)"*, which utilizes the Lottery Ticket Hypothesis (LTH) to identify an adaptive substructure within the dense networks that are tailored to the specific video input index. Our PFNR doesn't use memory, a pre-trained model, or fine-tuning for a sequential video representation compilation.

**Continual Learning.** Most continual learning approaches introduce extra memory like additional model capacity (Li et al., 2019; Yoon et al., 2020) or a replay buffer (Riemer et al., 2018; Chaudhry et al., 2019a; Buzzega et al., 2020; Arani et al., 2022; Sarfraz et al., 2023). However, several works have focused on building memory-efficient continual learners using pruning-based constraints to exploit initial model capability more compactly. CLNP (Golkar et al., 2019) selects important neurons for a given task using $\ell_1$ regularization to induce sparsity and freezes them to maintain performance. And pruned neurons are reinitialized for future task training. Piggyback (Mallya et al., 2018) trains task-specific binary masks on the weights given a pre-trained model. However, it does not allow for knowledge transfer among tasks, so the performance highly depends on the quality of the backbone model. HAT (Serrà et al., 2018) proposes task-specific learnable attention vectors to identify significant weights per task. The masks are formulated to layerwise cumulative attention vectors during continual learning. LL-Tickets (Chen et al., 2021b) recently suggests sparse subnetworks called lifelong tickets that perform well on all tasks during continual learning. The method searches for more prominent tickets from current ones if the obtained tickets cannot sufficiently learn the new task while maintaining performance on past tasks. However, LL-Tickets require external data to maximize knowledge distillation with learned models for prior tasks, and the ticket expansion process involves retraining and pruning steps. As a strong architecture-based baseline, WSN (Kang et al., 2022a) jointly learns the model weights and task-adaptive binary masks during continual learning. It prevents catastrophic forgetting of previous tasks by keeping the model weights selected, called winning tickets, intact at the end of each training. However, WSN is inappropriate for sequential complex video compilation since it reuses a few adaptive but sparse learnable parameters. To overcome the weakness of WSN, our PFNR explores more appropriate forget-free weights for representing complex video in Fourier space (Li et al., 2020a;b; Kovachki et al., 2021; Tran et al., 2021) using a newly proposed Fourier Subnetwork Operator (FSO).

## 3 PROGRESSIVE FOURIER NEURAL REPRESENTATION

This section presents our proposed continual neural implicit representation method, named *Progressive Fourier Neural Representation (PFNR)*. Given a supernet backbone, where we follow the NeRV (Chen et al., 2021a) architecture for video embedding and decoding, PFNR aims to expand

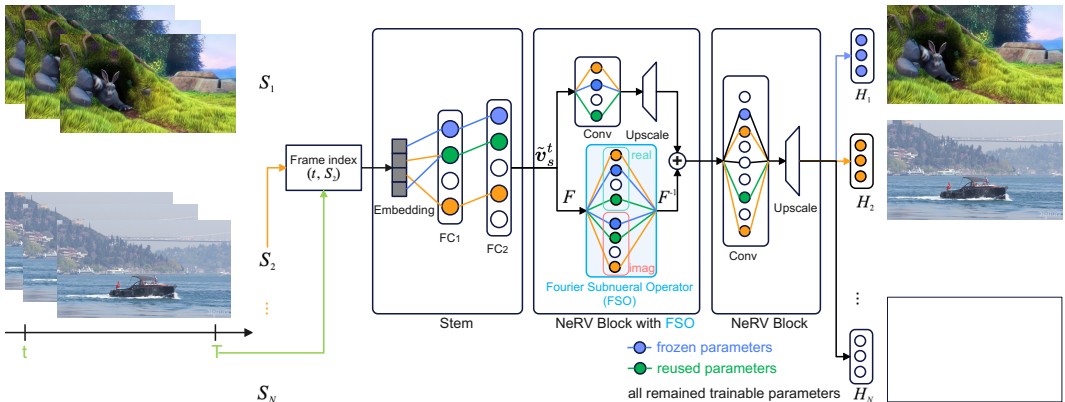

Figure 1: **Progressive Fourier Neural Representation (PFNR)**: PFNR takes time and video (session) indices as input and uses a sparse Stem + NeRV Blocks with *Fourier Subneural Operator* (FSO) to output the whole image through multi-heads $H_N$ where $\tilde{v}_s^t$ is a hidden representation. We denote frozen, reused, and trainable parameters in training at session 2. Note that each video representation is colored. In inference, we only need indices of session $s$ and frame $t$ and session mask (subnetwork).

its representation space continuously by sequentially encoding multiple videos within the Fourier space. As new videos arrive in the model, PFNR jointly updates the binary masks (including real and imaginary parts) with neural network weights, searching for the adaptive subnetwork to encode given videos. Once a video session is completed, we 'freeze' the weights of the chosen subnetwork. This approach ensures that the quality of previously learned representations and generated outputs remains unaffected by future training sessions, even if the new subnetwork structure shares some weights with videos encoded earlier. While the weights learned in earlier video sessions are frozen, we enable our PFNR to transfer prior knowledge to future video tasks (i.e., forward transfer). This makes the model adapt new videos effectively by leveraging the representation of past videos (Please see Figure 1).

**Problem Statement.** Let a video at $s_{th}$ session $\boldsymbol{V}_s = \{\boldsymbol{v}_t^s\}_{t=1}^{T_s} \in \mathbb{R}^{T_s \times H \times W \times 3}$ be represented by a function with the trainable parameter $\boldsymbol{\theta}$, $f_{\boldsymbol{\theta}} : \mathbb{R} \rightarrow \mathbb{R}^{H \times W \times 3}$, during Video Continual Learning (VCL), where $T_s$ denotes the number of frames in a video at session $s$, and $s \in \{1 \dots, |S|\}$. Given a session and frame index $s$ and $t$, respectively, the neural implicit representation aims to predict a corresponding RGB image $\boldsymbol{v}_t^s \in \mathbb{R}^{H \times W \times 3}$ by fitting an encoding function to a neural network: $\boldsymbol{v}_t^s = f_{\boldsymbol{\theta}}([s; t], H_s)$ where $H_s$ is $s_{th}$ head. For the sake of simplicity, we omit $H_s$ in the following equations. Let's consider a real-world learning scenario in which $|\mathcal{S}| = N$ or more sessions arrive in the model sequentially. We denote that $\mathcal{D}_s = \{\boldsymbol{e}_{s,t}, \boldsymbol{v}_{s,t}\}_{t=1}^{T_s}$ is the dataset of session $s$, composed of $T_s$ pairs of raw embeddings $\boldsymbol{e}_{s,t} = [\boldsymbol{e}_s; \boldsymbol{e}_t] \in \mathbb{R}^{1 \times 160}$ and corresponding frames $\boldsymbol{v}_t^s$. Here, we assume that $\mathcal{D}_s$ for session $s$ is only accessible when learning session $s$ due to the limited hardware memory and privacy-preserving issues, and session identity is given in the training and testing stages. The primary training objective in this sequence of $N$ video sessions is to minimize the following optimization problem:

$$\boldsymbol{\theta}^* = \underset{\boldsymbol{\theta}}{\text{minimize}} \frac{1}{N} \frac{1}{T_s} \sum_{s=1}^{N} \sum_{t=1}^{T_s} \mathcal{L}(f(\boldsymbol{e}_{s,t}; \boldsymbol{\theta}), \boldsymbol{v}_t^s), \qquad (1)$$

where the loss function $\mathcal{L}(\boldsymbol{v}_t^s)$ is composed of $\ell_1$ loss and *SSIM loss*. The former minimizes the pixel-wise RGB gap with the original input frames evenly, and the latter maximizes the similarity between the two entire frames based on luminance, contrast, and structure, as follows:

$$\mathcal{L}(\boldsymbol{V}_s) = \frac{1}{T_s} \sum_{t=1}^{T_s} \alpha ||\boldsymbol{v}_t^s - \hat{\boldsymbol{v}}_t^s||_1 + (1 - \alpha)(1 - \textbf{SSIM}(\boldsymbol{v}_t^s, \hat{\boldsymbol{v}}_t^s)), \qquad (2)$$

where $\hat{\boldsymbol{v}}_t^s$ is the output generated by the model $f$. For all experiments, we set the hyperparameter $\alpha$ to 0.7, and we adapt PixelShuffle (Shi et al., 2016) for session and time positional embedding.

Continual learners $f$ frequently use over-parameterized deep neural networks to ensure enough capacity for learning future tasks. This approach often leads to the discovery of subnetworks that perform as well as or better than the original network. Given the neural network parameters $\boldsymbol{\theta}$, the binary attention mask $\boldsymbol{m}_s^*$ that describes the optimal subnetwork for session $s$ such that $|\boldsymbol{m}_s^*|$ is less

than the model capacity $c$ follows as:

$$\boldsymbol{m}_s^* = \underset{\boldsymbol{m}_s \in \{0,1\}^{|\boldsymbol{\theta}|}}{\text{minimize}} \frac{1}{T_s} \sum_{t=1}^{T_s} \mathcal{L}\big(f(\boldsymbol{e}_{s,t}; \boldsymbol{\theta} \odot \boldsymbol{m}_s), \boldsymbol{v}_t^s\big) - \mathcal{J}, \quad \text{subject to } |\boldsymbol{m}_s^*| \le c, \qquad (3)$$

where session loss $\mathcal{J} = \mathcal{L}(\boldsymbol{v}_t^s)$ and $c \ll |\boldsymbol{\theta}|$ (used as the selected proportion % of model parameters in the following section). A robust model adhering to this condition is known as WSN (Kang et al., 2022a). However, WSN falls short in handling sequential complex video compilation due to its reliance on a limited set of adaptable yet sparse parameters in convolutional operators. In the following section, we introduce a novel Fourier Subnetwork Operator (FSO) to address this limitation.

## 3.1 FOURIER SUBNUERAL OPERATOR (FSO)

Conventional continual learner (i.e., WSN) only uses a few learnable parameters in convolutional operations to represent complex sequential image streams. To capture more parameter-efficient, forget-free NIRs, the NIR model requires fine discretization and video-specific sub-parameters. This motivation leads us to propose a novel subnetwork operator in Fourier space, which provides it with various bandwidths. Following the previous definition of Fourier convolutional operator (Li et al., 2020a), we adapt and redefine this definition to better fit the needs of the NIR framework. We use the symbol $\mathcal{F}$ to represent the Fourier transform of a function $f$, which maps from an embedding space of dimension $d_{\boldsymbol{e}} = 1 \times 160$ to a frame size denoted as $d_{\boldsymbol{v}}$. The inverse of this transformation is represented by $\mathcal{F}^{-1}$. In this context, we introduce our **F**ourier-integral **S**ubneural **O**perator (**FSO**), symbolized as $\mathcal{K}$, which is tailored to enhance the capabilities of our NIR system (see Appendix A.1):

$$\left(\mathcal{K}(\phi)\tilde{\boldsymbol{v}}_t^s\right)(\boldsymbol{e}_{s,t}) = \mathcal{F}^{-1}(R_{\boldsymbol{\phi}} \cdot (\mathcal{F}\tilde{\boldsymbol{v}}_t^s))(\boldsymbol{e}_{s,t}), \qquad (4)$$

where $\tilde{\boldsymbol{v}}_t^s$ is a hidden representation; $R_{\boldsymbol{\phi}}$ is the Fourier transform of a periodic subnetwork function which is parameterized by its subnetwork's parameters of real ($\boldsymbol{\theta}^{real} \odot \boldsymbol{m}_s^{real}$) and imaginary ($\boldsymbol{\theta}^{imag} \odot \boldsymbol{m}_s^{imag}$). We thus parameterize $R_{\boldsymbol{\phi}}$ separately as complex-valued tensors of real and imaginary $\boldsymbol{\phi}_{FSO} \in \{\boldsymbol{\theta}^{real}, \boldsymbol{\theta}^{imag}\}$. One key aspect of the FSO is that its parameters grow with the depth of the layer and the input/output size. However, through careful layer-wise inspection and adjustments for sparsity, we can find a balance that allows the FSO to describe neural implicit representations efficiently. In the experimental section, we will showcase the most efficient FSO structure and its performance. Figure 1 shows one possible PFNR structure of a single FSO. We describe the optimization in the following section.

---

**Algorithm 1** Progressive Fourier Neural Representation (PFNR) for VCL

---

    **input**: $\{\mathcal{D}_s\}_{s=1}^N$, model weights of FSO $\boldsymbol{\theta}_* = \{\boldsymbol{\theta}, \boldsymbol{\phi_{FSO}}\}$, score weights of FSO $\boldsymbol{\rho}_* = \{\boldsymbol{\rho}, \boldsymbol{\rho}_{FSO}\}$,
        binary mask $\mathbf{M}_0 = \{\mathbf{0}^{|\boldsymbol{\theta}|}, \mathbf{0}^{|\boldsymbol{\theta}_{FSO}|}\}$, and layer-wise capacity $c\%$.

1:  randomly initialize $\boldsymbol{\theta}_*$ and $\boldsymbol{\rho}_*$.
2:  **for** session $s = 1, \cdots, |\mathcal{S}|$ **do**
3:     **if** s > 1 **then**
4:         randomly re-initialize $\boldsymbol{\rho}_*$.
5:     **end if**
6:     **for** batch $\mathbf{b}_t \sim \mathcal{D}_s$ **do**
7:         obtain mask $\mathbf{m}_s$ of the top-$c\%$ scores $\boldsymbol{\rho}_*$ at each layer
8:         compute $\mathcal{L}\left(f(\boldsymbol{e}_{s,t}; \boldsymbol{\theta}_* \odot \mathbf{m}_s), \mathbf{b}_t\right)$, where input embedding, $\boldsymbol{e}_{s,t} = [\boldsymbol{e}_s; \boldsymbol{e}_t]$.
9:         $\boldsymbol{\theta}_* \leftarrow \boldsymbol{\theta}_* - \eta\left(\frac{\partial \mathcal{L}}{\partial \boldsymbol{\theta}_*} \odot (\mathbf{1} - \mathbf{M}_{s-1})\right)$                ▷ trainable weight update
10:      $\boldsymbol{\rho}_* \leftarrow \boldsymbol{\rho}_* - \eta(\frac{\partial \mathcal{L}}{\partial \boldsymbol{\rho}_*})$                                ▷ weight score update
11:     **end for**
12:     $\hat{\boldsymbol{\theta}}_s = \boldsymbol{\theta}_* \odot \mathbf{m}_s$
13:     $\mathbf{M}_s \leftarrow \mathbf{M}_{s-1} \vee \mathbf{m}_s$                             ▷ accumulate binary mask
14: **end for**
    **output**: $\{\hat{\boldsymbol{\theta}}_s\}_{s=1}^N$

---

## 3.2 SEQUENTIAL VIDEO REPRESENTATIONAL SUBNETWORKS

Let each weight $\boldsymbol{\theta}_* = \{\boldsymbol{\theta}, \boldsymbol{\phi}_{FSO}\}$ be associated with a learnable parameter we call *weight score* $\boldsymbol{\rho}_* = \{\boldsymbol{\rho}, \boldsymbol{\rho}_{FSO}\}$, which numerically determines the importance of the weight associated with it; that

is, a weight with a higher weight score is seen as more important. We find a sparse subnetwork $\hat{\boldsymbol{\theta}}_s$ of the neural network and assign it as a solver of the current session $s$. We use subnetworks instead of the dense network as solvers for two reasons: (1) Lottery Ticket Hypothesis (Frankle & Carbin, 2019) demonstrates the existence of a competitive subnetwork that is comparable with the dense network, and (2) the subnetwork requires less capacity than dense networks, and therefore it inherently reduces the size of the expansion of the solver.

Motivated by such benefits, we propose a novel PFNR, the joint-training method for sequential video representation compilation, as shown in Algorithm 1. The pseudo-code explains how to acquire subnetworks within a dense network. We find $\hat{\boldsymbol{\theta}}_s = \boldsymbol{\theta}_* \odot \boldsymbol{m}_s$ by selecting the top-$c$% weights from the weight scores $\boldsymbol{\rho}_*$, where $c$ is the target layer-wise capacity ratio in %; $\mathbf{m}_s$ is a session-dependent binary mask. Formally, $\mathbf{m}_s$ is obtained by applying a indicator function $\mathbb{1}_c$ on $\boldsymbol{\rho}_*$ where $\mathbb{1}_c(\rho_*) = 1$ if $\rho_*$ belongs to top-$c$% scores and 0 otherwise. Therefore, the subnetworks $\{\hat{\boldsymbol{\theta}}_s\}_{s=1}^N$ for all video session $\mathcal{S}$ are obtained by $\hat{\boldsymbol{\theta}}_s = \boldsymbol{\theta}_* \odot \mathbf{m}_s$. Straight-through estimator (Bengio et al., 2013; Hinton, 2012; Ramanujan et al., 2020) is used to update $\boldsymbol{\rho}_*$, which ignores the derivative of the indicator function and passes on the incoming gradient as if the indicator function were an identity function.

## 4 EXPERIMENTS

Our method is validated on benchmark datasets for Video Task-incremental Learning (VTL) and compared against various continual learning baselines. In all experiments conducted for this paper, we utilize a multi-head configuration for continual video representation learning. This means the session identifier, denoted as $s$, is provided during the training and inference phases. Our experimental setups align with NeRV (Chen et al., 2021a) and HNeRV (Chen et al., 2023).

**Datasets.** *1) UVG of 8 Video Sessions*: We experiment on eight sequential videos to validate our PFNR. The eight videos consist of one from the scikit-video and seven from the UVG dataset. The category index and order in UVG8 are as follows: *1.bunny, 2.beauty, 3.bosphorus, 4.bee, 5.jockey, 6.setgo, 7.shake, 8.yacht*.

*2) UVG of 17 Video Sessions*: We conducted an extended experiment on 17 video sessions by adding 9 more videos to the UVG of 8 video sessions. The category index and order in UVG17 are as follows: *1.bunny, 2.city, 3.beauty, 4.focus, 5.bosphorus, 6.kids, 7.bee, 8.pan, 9.jockey, 10.lips, 11.setgo, 12.race, 13.shake, 14.river, 15.yacht, 16.sunbath, 17.twilight*. Please refer to the supplementary material.

**Architecture.** We employ NeRV as our baseline architecture and follow its details for a fair comparison. After the positional encoding, we apply 2 sparse MLP layers on the output of the positional encoding layer, followed by five sparse NeRV blocks with upscale factors of 5, 2, 2, 2, 2. These sparse NeRV blocks decode 1280×720 frames from the 16×9 feature map obtained after the sparse MLP layers. For the upscaling method in the sparse NeRV blocks, we also adopt PixelShuffle (Shi et al., 2016). Fourier Subneural Operator (FSO) is used at the NeRV2 or NeRV3 layer, as depicted in Table 7. The positional encoding for the video index $s$ and frame index $t$ is as follows:

$$\boldsymbol{\Gamma}(s,t) = [\ \sin(b^0\pi s), \cos(b^0\pi s), \cdots, \sin(b^{l-1}\pi s), \cos(b^{l-1}\pi s),$$
$$\sin(b^0\pi t), \cos(b^0\pi t), \cdots, \sin(b^{l-1}\pi t), \cos(b^{l-1}\pi t)\ ], \tag{5}$$

where the hyperparameters are set to $b = 1.25$ and $l = 80$ such that $\boldsymbol{\Gamma}(s,t) \in \mathbb{R}^{1 \times 160}$. As differences from the previous NeRV model, the first layer of the MLP has its input size expanded from 80 to 160 to incorporate both frame and video indices, and distinct head layers after the NeRV block are utilized for each video. For the loss objective in Equation 2, $\alpha$ is set to 0.7. We evaluate the video quality, average video session quality, and backward transfer with two metrics: PSNR and MS-SSIM (Wang et al., 2003). We implement our model in PyTorch and train it in full precision (FP32). All experiments are run with NVIDIA RTX8000.

**Baselines.** To show the effectiveness, we compare our PFNR with strong CL baselines: Single-Task Learning (STL), which trains on single tasks independently, EWC (Kirkpatrick et al., 2017), which is a regularized baseline, iCaRL (Rebuffi et al., 2017), and ESMER (Sarfraz et al., 2023) which are current strong rehearsal-based baseline, WSN (Kang et al., 2022a) which is a current strong architecture-based baseline, and Multi-Task Learning (MTL) which trains on multiple video sessions simultaneously, showing the upper-bound of WSN. Except for STL, all models are trained and evaluated on multi-head settings where a video session and time $(s, t)$ indices are provided.

Table 1: PSNR results with Fourier Subnueral Operator (FSO) layer ($f$-NeRV$*$) (detailed in Table 7) on UVG8 Video Sessions with average PSNR and Backward Transfer (BWT). Note that $*$ denotes our reproduced results.

| Method | Video Sessions | | | | | | | | Avg. PSNR / BWT |
|---|---|---|---|---|---|---|---|---|---|
| | 1 | 2 | 3 | 4 | 5 | 6 | 7 | 8 | |
| STL, NeRV Chen et al. (2021a)$*$ | 39.66 | 36.28 | 38.14 | 42.03 | 36.58 | 29.22 | 37.27 | 31.45 | 36.33 / - |
| EWC Kirkpatrick et al. (2017)$*$ | 10.19 | 11.15 | 14.47 | 8.39 | 12.21 | 10.27 | 9.97 | 23.98 | 12.58 / -17.59 |
| iCaRL Rebuffi et al. (2017)$*$ | 30.84 | 26.30 | 27.28 | 34.48 | 20.90 | 17.28 | 30.33 | 24.64 | 26.51 / -3.90 |
| ESMER Sarfraz et al. (2023)$*$ | 31.71 | 23.09 | 24.15 | 28.03 | 17.30 | 13.81 | 12.45 | 24.57 | 21.92 / -9.99 |
| WSN$*$, c = 50.0 % | 34.05 | 32.28 | 29.98 | 32.88 | 22.15 | 18.61 | 27.68 | 23.64 | 27.66 / 0.0 |
| PFNR , c = 50.0 %, $f$-NeRV2 | 34.46 | 33.91 | 32.17 | 36.43 | 25.26 | 20.74 | 30.18 | 25.45 | 29.82 / 0.0 |
| PFNR , c = 50.0 %, $f$-NeRV3 | **36.45** | **35.15** | **35.10** | **38.57** | **28.07** | **23.06** | **32.83** | **27.70** | **32.12 / 0.0** |
| MTL (upper-bound) | 34.22 | 32.79 | 32.34 | 38.33 | 25.30 | 22.44 | 33.73 | 27.05 | 30.78 / - |

Table 2: MS-SSIM results with Fourier Subnueral Operator (FSO) layer ($f$-NeRV$*$) (detailed in Table 7) on UVG8 Video Sessions with average MS-SSIM, Backward Transfer (BTW) of MS-SSIM. Note that $*$ denotes our reproduced results.

| Method | Video Sessions | | | | | | | | Avg. MS-SSIM / BWT |
|---|---|---|---|---|---|---|---|---|---|
| | 1 | 2 | 3 | 4 | 5 | 6 | 7 | 8 | |
| STL, NeRV Chen et al. (2021a)$*$ | 0.99 | 0.95 | 0.98 | 0.99 | 0.97 | 0.96 | 0.98 | 0.96 | 0.97 / - |
| EWC Kirkpatrick et al. (2017)$*$ | 0.22 | 0.23 | 0.35 | 0.10 | 0.27 | 0.19 | 0.21 | 0.79 | 0.30 / -0.62 |
| iCaRL Rebuffi et al. (2017)$*$ | 0.94 | 0.80 | 0.82 | 0.97 | 0.59 | 0.57 | 0.92 | 0.81 | 0.80 / -0.11 |
| ESMER Sarfraz et al. (2023)$*$ | 0.88 | 0.65 | 0.68 | 0.90 | 0.42 | 0.32 | 0.19 | 0.81 | 0.61 / -0.33 |
| WSN$*$, c = 50.0 % | 0.98 | 0.91 | 0.90 | 0.97 | 0.74 | 0.62 | 0.88 | 0.77 | 0.85 / 0.0 |
| PFNR , c = 50.0 %, $f$-NeRV2 | 0.98 | 0.93 | 0.93 | 0.99 | 0.83 | 0.75 | 0.92 | 0.84 | 0.90 / 0.0 |
| PFNR , c = 50.0 %, $f$-NeRV3 | **0.99** | **0.94** | **0.97** | **0.99** | **0.88** | **0.84** | **0.95** | **0.90** | **0.93 / 0.0** |
| MTL (upper-bound) | 0.98 | 0.91 | 0.93 | 0.99 | 0.84 | 0.82 | 0.95 | 0.89 | 0.91 / - |

**Training.** In all experiments, we follow the same experimental settings as NeRV (Chen et al., 2023) and HNeRV (Chen et al., 2023) for fair comparisons. We train WSN, PFNR, NeRV (STL), and MTL using Adam optimizer with a learning rate 5e-4. For the ablation study on UVG8 and UVG17, we use a cosine annealing learning rate schedule (Loshchilov & Hutter, 2016), batch size of 1, training epochs of 150, and warmup epochs of 30 unless otherwise denoted.

**VCL's performance metrics of PSNR & MS-SSIM.** We evaluate all methods based on the following continual learning metrics:

1. *Average PSNR or MS-SSIM (i.e., Ave. PSNR)* measures the average of the final performances on all video sessions: PSNR or $\text{MS-}SSIM = \frac{1}{N} \sum_{s=1}^{N} A_{N,s}$, where $A_{N,s}$ is the test PSNR or MS-SSIM for session $s$ after training on the final video session $S$.

2. *Backward Transfer (BWT) of PSNR or MS-SSIM* measures the video representation forgetting during continual learning. Negative BWT means that learning new video sessions causes the video representation forgetting of past sessions: $\text{BWT} = \frac{1}{N-1} \sum_{s=1}^{N-1} A_{N,s} - A_{s,s}$.

Table 3: PSNR results with Fourier Subnueral Operator (FSO) layer ($f$-NeRV$*$) (detailed in Table 7) on UVG17 Video Sessions with average PSNR and Backward Transfer (BWT) of PSNR. Note that $*$ denotes our reproduced results.

| Method | Video Sessions | | | | | | | | | | | | | | | | | Avg. PSNR BWT |
|---|---|---|---|---|---|---|---|---|---|---|---|---|---|---|---|---|---|---|
| | 1 | 2 | 3 | 4 | 5 | 6 | 7 | 8 | 9 | 10 | 11 | 12 | 13 | 14 | 15 | 16 | 17 | |
| STL, NeRV Chen et al. (2021a)$*$ | 39.66 | 44.89 | 36.28 | 41.13 | 38.14 | 31.53 | 42.03 | 34.74 | 36.58 | 36.85 | 29.22 | 31.81 | 37.27 | 34.18 | 31.45 | 38.41 | 43.86 | 36.94 / - |
| EWC Kirkpatrick et al. (2017)$*$ | 11.15 | 9.21 | 12.71 | 11.40 | 15.58 | 9.25 | 7.06 | 12.96 | 6.34 | 10.31 | 9.55 | 13.39 | 5.76 | 8.67 | 10.93 | 10.92 | 28.29 | 11.38 / -16.13 |
| iCaRL Rebuffi et al. (2017)$*$ | 24.31 | 28.25 | 22.19 | 22.74 | 22.84 | 16.55 | 29.37 | 17.92 | 16.65 | 27.43 | 13.64 | 16.42 | 24.02 | 21.60 | 19.40 | 18.60 | 26.46 | 21.67 / -6.23 |
| ESMER Sarfraz et al. (2023)$*$ | 30.77 | 26.33 | 22.79 | 21.35 | 23.76 | 13.64 | 28.25 | 15.22 | 16.71 | 23.78 | 13.35 | 15.23 | 18.21 | 19.22 | 24.59 | 20.61 | 22.42 | 20.95 / -15.23 |
| WSN$*$, c = 30.0 % | 31.50 | 34.37 | 31.00 | 32.38 | 29.26 | 23.08 | 31.96 | 22.64 | 22.07 | 33.48 | 18.34 | 20.45 | 27.21 | 24.33 | 23.09 | 21.23 | 29.13 | 26.80 / 0.0 |
| PFNR , c = 30.0 %, $f$-NeRV2 | 32.01 | 35.84 | 32.97 | 35.17 | 31.24 | 24.82 | 36.01 | 25.85 | 24.83 | 35.76 | 20.50 | 22.79 | 30.40 | 27.37 | 25.52 | 25.40 | 32.70 | 29.36 / 0.0 |
| PFNR , c = 30.0 %, $f$-NeRV3 | **33.64** | **39.24** | **34.21** | **37.79** | **34.05** | **27.17** | **38.17** | **29.79** | **26.56** | **36.18** | **22.97** | **24.36** | **32.50** | **30.22** | **27.62** | **29.15** | **35.68** | **31.72 / 0.0** |
| MTL (upper-bound) | 32.39 | 34.35 | 31.45 | 34.03 | 30.70 | 24.53 | 37.13 | 27.83 | 23.80 | 34.69 | 20.77 | 22.37 | 32.71 | 28.00 | 25.89 | 26.40 | 33.16 | 29.42 / - |

## 4.1 COMPARISONS WITH BASELINES

**PSNR & MS-SSIM.** To compare PFNR with conventional representative continual learning methods such as EWC, iCaRL, ESMER, and WSN, we prepare the reproduced results, as shown in Table 1, 2, 3, and 4. The architecture-based WSN outperformed the regularized method and replay method. The sparseness of WSN does not significantly affect sequential video representation results on two sequential benchmark datasets. Our PFNR outperforms all conventional baselines including WSN and MLT (upper-bound of WSN) on the UVG8 and UVG17 benchmark datasets. Moreover, our performances of PFNR with $f$-NeRV3 are better than those of $f$-NeRV2 since $f$-NeRV3 tends to represent local textures, stated in the following Section 4.2. Note that the number of parameters of MLT is precisely the same as those of WSN.

Table 4: MS-SSIM results with Fourier Subnueral Operator (FSO) layer ($f$-NeRV$*$) (detailed in Table 7) on UVG17 Video Sessions with average MS-SSIM and Backward Transfer (BWT) of MS-SSIM. Note that $*$ denotes our reproduced results.

| Method | \multicolumn{17}{c}{Video Sessions} | Avg. MS-SSIM BWT |
|---|---|---|---|---|---|---|---|---|---|---|---|---|---|---|---|---|---|---|
| | 1 | 2 | 3 | 4 | 5 | 6 | 7 | 8 | 9 | 10 | 11 | 12 | 13 | 14 | 15 | 16 | 17 | |
| STL, NeRV Chen et al. (2021a)* | 0.99 | 0.99 | 0.95 | 0.98 | 0.98 | 0.96 | 0.99 | 0.98 | 0.97 | 0.95 | 0.96 | 0.96 | 0.98 | 0.98 | 0.96 | 0.99 | 0.99 | 0.97 / - |
| EWC Kirkpatrick et al. (2017)* | 0.26 | 0.24 | 0.44 | 0.24 | 0.40 | 0.29 | 0.15 | 0.17 | 0.26 | 0.26 | 0.17 | 0.34 | 0.04 | 0.30 | 0.33 | 0.31 | 0.91 | 0.30 / -0.55 |
| iCaRL Rebuffi et al. (2017)* | 0.74 | 0.88 | 0.67 | 0.67 | 0.64 | 0.48 | 0.91 | 0.53 | 0.37 | 0.82 | 0.35 | 0.53 | 0.75 | 0.70 | 0.61 | 0.60 | 0.87 | 0.65 / -0.20 |
| ESMER Sarfraz et al. (2023)* | 0.85 | 0.86 | 0.64 | 0.63 | 0.66 | 0.46 | 0.89 | 0.51 | 0.42 | 0.79 | 0.30 | 0.51 | 0.43 | 0.68 | 0.82 | 0.6 | 0.63 | 0.62 / -0.37 |
| WSN*, c = 30.0 % | 0.96 | 0.97 | 0.89 | 0.93 | 0.88 | 0.77 | 0.97 | 0.77 | 0.73 | 0.91 | 0.60 | 0.74 | 0.86 | 0.81 | 0.76 | 0.72 | 0.93 | 0.84 / 0.0 |
| PFNR , c = 30.0 %, $f$-NeRV2 | 0.97 | 0.98 | 0.92 | 0.95 | 0.92 | 0.84 | 0.98 | 0.88 | 0.82 | 0.93 | 0.73 | 0.82 | 0.92 | 0.89 | 0.84 | 0.87 | 0.97 | 0.90 / 0.0 |
| PFNR , c = 30.0 %, $f$-NeRV3 | **0.98** | **0.99** | **0.93** | **0.97** | **0.96** | **0.91** | **0.99** | **0.96** | **0.87** | **0.94** | **0.84** | **0.87** | **0.94** | **0.94** | **0.90** | **0.94** | **0.98** | **0.94 / 0.0** |
| MTL (upper-bound) | 0.97 | 0.97 | 0.90 | 0.94 | 0.91 | 0.82 | 0.99 | 0.92 | 0.80 | 0.92 | 0.75 | 0.81 | 0.94 | 0.90 | 0.85 | 0.89 | 0.97 | 0.90 / - |

**PFNR's Compression.** We follow NeRV's video quantization and compression pipeline (Chen et al., 2021a), except for the model pruning step, to evaluate performance drops and backward transfer in the video sequential learning, as shown in Figure 2. Once sequential training is done, our PFNR doesn't need any extra prune and finetune steps, unlike NeRV. This point is our key advantage of PFNR over NeRV. Figure 2 (a) shows the results of various sparsity and bit-quantization on the UVG17 datasets: the 8bit PFNR's performances are comparable with 32bit ones without a significant video quality drop. From our observations, the 8-bit subnetwork seems to be enough for video implicit representation. Figure 2 (b) shows the rate-distortion curves. We compare PFNR with WSN and NeRV (STL). For a fair comparison, we take steps of pruning, fine-tuning, quantizing, and encoding NeRV. Our PFNR outperforms all baselines.

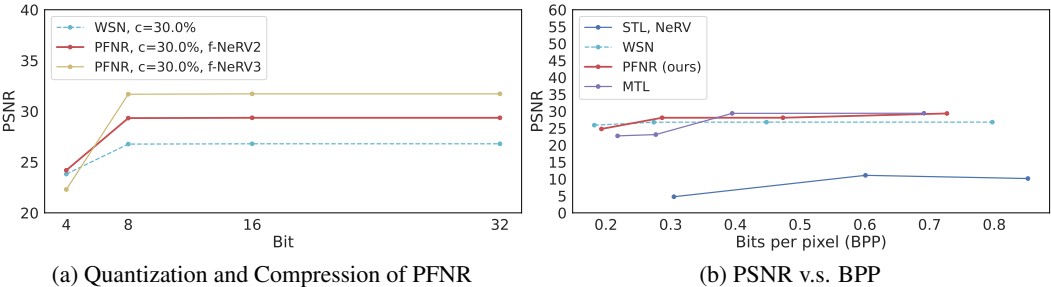

(a) Quantization and Compression of PFNR  (b) PSNR v.s. BPP

Figure 2: PSNR v.s. Bits-per-pixel (BPP) on the UVG17 datasets

**Performance and Capacity.** Our PFNR outperforms WSN and MTL, as stated in Figure 3 (a). This result might suggest that properly selected weights in Fourier space lead to generalization more than others in VCL. Moreover, to show the behavior of PSNR, We prepare a progressive PSNR's capacity and investigate how PFNR reuses weights over sequential video sessions, as shown in Figure 3 (b). PFNR tends to progressively transfer weights used for a prior session to weights for new ones, but the proposition of reused weights gets smaller as video sessions increase.

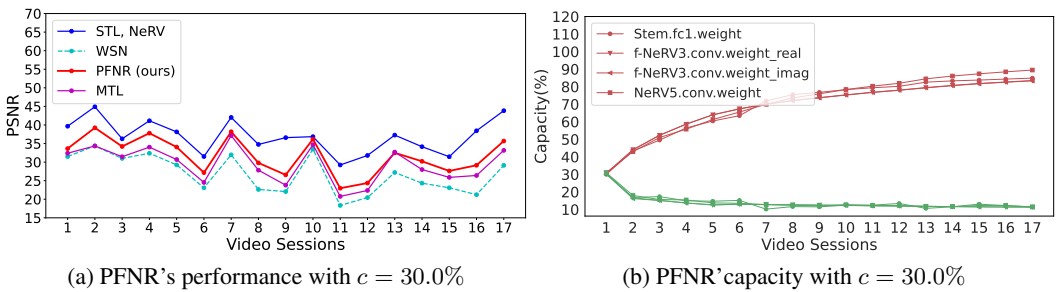

(a) PFNR's performance with $c = 30.0\%$  (b) PFNR'capacity with $c = 30.0\%$

Figure 3: PFNR's Comparison of PSNR with others and layer-wise accumulated capacities on the UVG17 dataset. Note that, in (b), green represents the percentage of reused subnetwork's parameters of Stem, $f$-NeRV3, and NeRV5 at the current session (s) obtained at the past (s-1) video sessions

## 4.2 PFNR'S VIDEO REPRESENTATIONS

We prepare the results of video generation as shown in Figure 5. We demonstrate that a sparse solution (PFNR with $c = 30.0\%$, $f$-NeRV3) generates video representations sequentially without significant performance drops. Compared with WSN, PFNR provides more precise representations.

To find out the results, we inspect the layer-wise representations as shown in Figure 4, which provides essential observations that PFNR tends to capture local textures broadly at the NeRV3 layer while WSN focuses on local objects. This behavior of PFNR could lead to more generalized performances. Moreover, we conduct an ablation study to inspect the best sparsity of $f$-NeRV3 while holding the remaining parameters' sparsity (c=50.0 %), as shown in Figure 10. Please refer to the supplementary materials for comparisons with baselines.

| NeRV3 | NeRV4 | NeRV5 | NeRV6 | Head |
| --- | --- | --- | --- | --- |

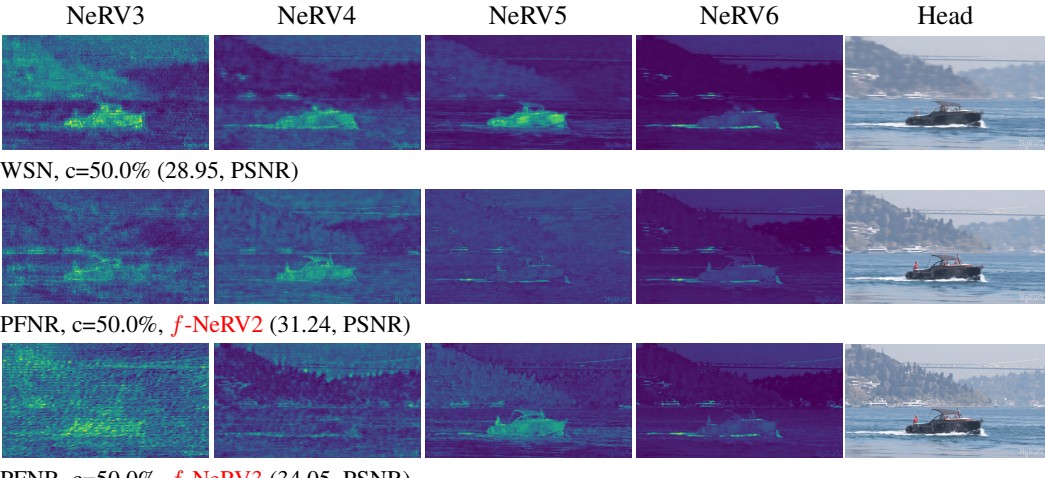

WSN, c=50.0% (28.95, PSNR)

PFNR, c=50.0%, $f$-NeRV2 (31.24, PSNR)

PFNR, c=50.0%, $f$-NeRV3 (34.05, PSNR)

Figure 4: PFNR's Representations of NeRV Blocks with $c = 50.0\%$ on the UVG17 dataset.

| t=0 | t=1 | t=2 | t=3 |
| --- | --- | --- | --- |

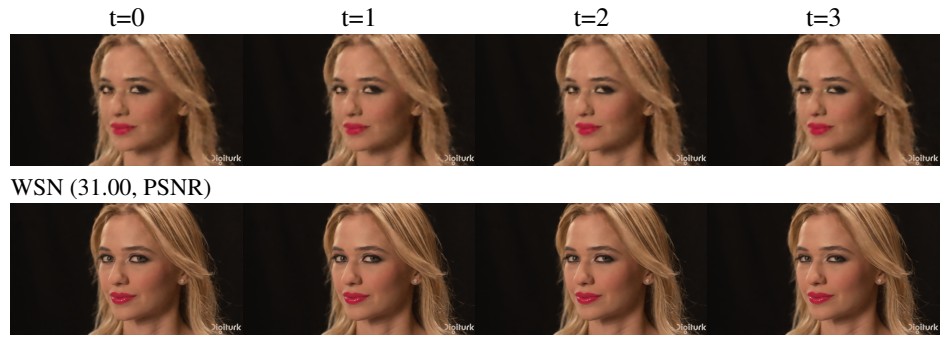

WSN (31.00, PSNR)

PFNR, $f$-NeRV3 (34.21, PSNR)

Figure 5: PFNR's Video Generation (from t=0 to t=3) with $c = 30.0\%$ on the UVG17 dataset.

## 5 CONCLUSION

Neural Implicit Representations (NIR) have gained significant attention recently due to their ability to represent complex and high-dimensional data. Unlike explicit representations, which require storing and manipulating individual data points, implicit representations capture information through a learned mapping function without explicitly representing the data points themselves. While they often compress neural networks substantially to accelerate encoding/decoding speed, yet existing methods fail to transfer learned representations to new videos. This work investigates the continuous expansion of implicit video representations as videos arrive sequentially over time, where the model can only access the videos from the current session. To tackle this problem, we propose a novel neural video representation, *Progressive Fourier Neural Representation (PFNR)*, that finds an adaptive substructure from the supernet to the given video based on Lottery Ticket Hypothesis in a complex domain. At each training session, our PFNR transfers the learned knowledge of the previously obtained subnetworks to obtain the representation of the current video without modifying past subnetwork weights. Therefore, it can perfectly preserve the decoding ability (i.e., catastrophic forgetting) on previous videos. We demonstrate the effectiveness of our proposed PFNR over baselines on the novel UVG8/17 and DAVIS50 video sequence benchmark datasets.

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

## A  APPENDIX

### A.1  FOURIER SUBNEURAL OPERATOR (FSO)

To elucidate the Fourier Subneural Operator (FSO) as delineated in Equation 4, We'll delve into a comprehensive review of the Neural (Li et al., 2020b) Operator and Fourier Operator (Li et al., 2020a), focusing on its discretization methodology.

#### A.1.1  DISCRETIZATION FOR NEURAL/FOURIER OPERATORS

Let's consider $D_j = \{x_1, \cdots, x_n\} \subset D$ as an $n$-point discretization of the domain $D$. Given this setting, we have observation $a_{j|D_j} \in \mathbb{R}^{n \times d_a}$, $u_{j|D_j} \in \mathbb{R}^{n \times d_v}$, pertaining to a finite collection of input-output pairs indexed by $j$. The goal of achieving discretization-invariance with a neural operator implies that the operator is capable of generating a response $u(x)$ for any point $x$ in the domain $D$, even for instances where $x$ may not be an element of the discretized subset $D_j$. This characteristic ensures the neural operator maintains its predictive and functional integrity across the continuous domain $D$, notwithstanding the specific discretization points representing $D$.

#### A.1.2  NEURAL OPERATOR

The neural operator, as described by Li et al. (2020b) is formulated as an iterative architecture denoted by $v_0 \mapsto v_1 \mapsto, \cdots, \mapsto v_T$ where $v_j$ (for $j = 0, 1, \cdots, T - 1$) represents a sequence of functions. Each function in this sequence yields values in the space $\mathbb{R}^{d_v}$. As the process iterates, the transformation from one state $v_t$ to the next state $v_{t+1}$ is defined by the interplay of two distinct

types of operations: a non-local integral operator $\mathcal{K}$ and a local, nonlinear activation function $\sigma$. Specifically, the update from $v_t$ to $v_{t+1}$ in each iteration is articulated as the composition of these two operations, mathematically represented as follows:

$$v_{t+1}(x) := \sigma(Wv_t(x) + (\mathcal{K}(a;\phi)v_t(x)), \quad \forall x \in D \tag{6}$$

In this formulation, a functional operator $\mathcal{K} : \mathcal{A} \times \Theta_{\mathcal{K}} \rightarrow \mathcal{L}(\mathcal{U}(D;\mathbb{R}^{d_v}), \mathcal{U}(D;\mathbb{R}^{d_v}))$ maps to bounded linear operators on the function space $\mathcal{U}(D;\mathbb{R}^{d_v})$. This mapping is parameterized by $\phi \in \Theta_{\mathcal{K}}$. Additionally, the function $W : \mathbb{R}^{d_v} \rightarrow \mathbb{R}^{d_v}$ is a linear transformation, and $\sigma : \mathbb{R} \rightarrow \mathbb{R}$ is a non-linear activation function whose action is defined-component-wise.

For increasing integration in defining complex, flexible functional mappings, Li et al. (2020a) choose $\mathcal{K}(a;\phi)$ to be a kernel integral transformation parameterized by a neural network. The kernel integral operator mapping in Equation 6 is defined by

$$(\mathcal{K}(a;\phi)v_t)(x) := \int_D k(x, y, a(x), a(y); \phi)v_t(y)dy, \quad \forall x \in D \tag{7}$$

where $k_\phi : \mathbb{R}^{2(d+d_a)} \rightarrow \mathbb{R}^{d_v \times d_v}$ is a neural network parameterized by $\phi \in \Theta_{\mathcal{K}}$. Here, $k_\phi$ plays the role of a kernel function, which we learn from data. Together Equation 6 and Equation 7 constitute a generalization of neural networks to infinite-dimensional spaces.

By removing the dependence on the function $a$ and enforcing a shift-invariance property in the kernel function $k_\phi(x, y) = k_\phi(x - y)$, the operator simplifies into a convolution operator. This transformation aligns the kernel integral operator with the principles of fundamental solutions and leverages the intrinsic properties of convolution operations.

$$(\mathcal{K}(a;\phi)v_t)(x) := \int_D k_\phi(x - y)v_t(y)dy, \quad \forall x \in D \tag{8}$$

This fact by parameterizing $K_\phi$ directly is exploited in Fourier space and used in the Fast Fourier Transform (FFT) to compute Equation 8 efficiently.

### A.1.3 FOURIER OPERATOR

Li et al. (2020a) suggest replacing the kernel integral operator in Equation 7, by a convolution operator (see Equation 8) defined in Fourier space. Let $\mathcal{F}$ denote the Fourier transform of a function $f : D \rightarrow \mathbb{R}^{d_v}$ and $\mathcal{F}^{-1}$ its inverse then

$$(\mathcal{F}f)_j(k) = \int_D f_j(x)e^{-2i\pi<x,k>}dx, \quad (\mathcal{F}^{-1}f)_j(x) = \int_D f_j(k)e^{2i\pi<x,k>}dk$$

for $j = 1, \cdots, d_v$ where $i = \sqrt{-1}$ is the imaginary unit. By letting $k_\phi(x, y, a(x), a(y)) = k_\phi(x - y)$ in Equation 8 and applying the convolution theorem, the convolutional operation in Fourier space follows as:

$$(\mathcal{K}(a;\phi)v_t)(x) = \mathcal{F}^{-1}(\mathcal{F}(k_\phi) \cdot \mathcal{F}(v_t))(x), \quad \forall \in D.$$

Furthermore, the Fourier integral operator is defined directly by parameterizing $k_\phi$ in Fourier space as follows:

$$(\mathcal{K}(\phi)v_t)(x) = \mathcal{F}^{-1}(R_\phi \cdot (\mathcal{F}v_t))(x) \quad \forall x \in D \tag{9}$$

where $R_\phi$ is the Fourier transform of a periodic function (i.e., sine and cosine functions in the time domain) $k : \bar{D} \rightarrow \mathbb{R}^{d_v \times d_v}$ parameterized by $\phi \in \Theta_{\mathcal{K}}$.

For a given frequency mode $k \in D$, the Fourier transformation of $v_t$, denoted as $(\mathcal{F}v_t)(k) \in \mathbb{C}^{d_v}$. Similarly, $R_\phi(k) \in \mathbb{C}^{d_v \times d_v}$, representing a complex-valued matrix associated with each frequency mode $k$. Key aspects of this setup include:

- (Periodicity and Fourier Series Expansion): Given the periodic nature of $k$, it can be represented by a Fourier series. This allows for the analysis and computation to be conducted in terms of discrete modes $k \in \mathbb{Z}^d$.

- (Truncation and Finite-dimensional Parameterization): To manage the complexity and ensure computational feasibility, the Fourier series is truncated at a maximal number of modes $k_{max}$. The truncation is quantified by the set $Z_{k_{max}}$, which includes all modes $k \in \mathbb{Z}^d$ that satisfy the condition $|k_j| \leq k_{max,j}$ for each dimension $j = 1, \cdots, d$. The operator $R_\phi$ is parameterized as a complex-valued tensor of shape $(k_{max} \times d_v \times d_v)$, comprising the collection of truncated Fourier modes.

- (Conjugate Symmetry and Real-value of $k$): Due to the real-valued nature of $k$, conjugate symmetry is imposed on the Fourier coefficients. This is a fundamental property of the Fourier transform of real-valued functions, ensuring that the resulting inverse Fourier transform yields a real-valued function.

- (Choice of $Z_{k_{max}}$ and Efficiency Considerations): While the canonical choice for low-frequency modes typically involves an upper bound on the $\ell_1$-norm of $k \in \mathbb{Z}^d$, the set $Z_{k_{max}}$ is in this work chosen based on different criteria for efficient implementation.

### A.1.4 FOURIER SUBNUERAL OPERATOR (FSO)

(Discretization for VIL): One difference exists between prior physical modeling for Neural/Fourier Operators and neural implicit representation for Fourier Subneural Operator (FSO). Following the previous definition (Li et al., 2020a), $v_t$ is the temporal length. In the VIL setting, the function learns a time-specific continuous hidden output (an implicit representation, $\tilde{v}_t^s$) given discrete session and time indices (session $s$, time, $t$).

(Fourier Subneural Opeartor (FSO): Conventional continual learner (i.e., WSN) only uses a few learnable parameters in convolutional operations to represent complex sequential image streams. To capture more parameter-efficient, forget-free NIRs, the NIR model requires fine discretization and video-specific sub-parameters. This motivation leads us to propose a novel subnetwork operator in Fourier space, which provides it with various bandwidths. Following the previous definition of Fourier convolutional operator (Li et al., 2020a), we adapt and redefine this definition to better fit the needs of the NIR framework. We use the symbol $\mathcal{F}$ to represent the Fourier transform of a function $f$, which maps from an embedding space of dimension $d_e = 1 \times 160$ to a frame size denoted as $d_v$. The inverse of this transformation is represented by $\mathcal{F}^{-1}$. In this context, we introduce our **F**ourier-integral **S**ubneural **O**perator (**FSO**), symbolized as $\mathcal{K}$, which is tailored to enhance the capabilities of our NIR system:

$$(\mathcal{K}(\phi)\tilde{v}_t^s)(e_{s,t}) = \mathcal{F}^{-1}(R_\phi \cdot (\mathcal{F}\tilde{v}_t^s))(e_{s,t}), \tag{10}$$

where $\tilde{v}_t^s$ is a hidden representation, as shown in Figure 1; $R_\phi$ is the Fourier transform of a periodic subnetwork function, $k_\phi$ stated in Equation 8 which is parameterized by its subnetwork's parameters of real ($\theta^{real} \odot m_s^{real}$) and imaginary ($\theta^{imag} \odot m_s^{imag}$). We thus parameterize $R_\phi$ separately as complex-valued tensors of real and imaginary $\phi_{FSO} \in \{\theta^{real}, \theta^{imag}\}$. One key aspect of the FSO is that its parameters grow with the depth of the layer and the input/output size since the operator $R_\phi$, as stated in Equation 9, is parameterized as a complex-valued tensor of shape ($k_{max} \times d_{\tilde{v}} \times d_{\tilde{v}}$). However, through careful layer-wise inspection and adjustments for sparsity, we can find a balance that allows the FSO to describe neural implicit representations efficiently. In the experimental section, we will showcase the most efficient FSO structure and its performance. Figure 1 shows one possible PFNR structure of a single FSO.

### A.2 DATASETS

*1) UVG of 8 Video Sessions*: For "Big Buck Bunny" frames collected from the scikit-video, we use the frames provided with the NeRV official code. "Big Buck Bunny" comprises 132 frames of $720 \times 1080$ resolution. The frames for the other seven videos, collected from the UVG dataset, are extracted from YUV Y4M videos, and further information can be found in the implementation details. As shown in Table 5, the seven videos have $1920 \times 1080$ resolution, with the shaking video comprising 300 frames and the other 6 videos containing 600 frames each. These videos are captured at 120 frames per second (FPS), and the duration of the shaking video is 2.5 seconds, while the duration of the other 6 videos is 5 seconds. For convenience, the video titles in the UVG of 8 Video Sessions are abbreviated, and their corresponding full titles in the UVG dataset are as follows: *1.bunny : Big Buck Bunny, 2.beauty : Beauty, 3.bosphorus : Bosphorus, 4.bee : HoneyBee, 5.jockey : Jockey, 6.setgo : ReadySetGo, 7.shake : ShakeNDry, 8.yacht : YachtRide.*

*2) UVG of 17 Video Sessions*: Compared to the UVG of 8 video sessions, the other nine videos are all collected from the UVG dataset. The frames for these videos are extracted from YUV RAW videos with a resolution of 1920x1080. Further information can be found in the implementation details. As shown in Table 6, the sunbath video consists of 300 frames at 50 FPS and 12 seconds, the lips video consists of 600 frames at 120 FPS and 5 seconds, and the other seven videos comprised of 600 frames

Table 5: the UVG8 Video Sessions.

| | Video Sessions | | | | | | | |
|---|---|---|---|---|---|---|---|---|
| | 1 | 2 | 3 | 4 | 5 | 6 | 7 | 8 |
| Categories | bunny | beauty | bosphorus | bee | jockey | setgo | shake | yacht |
| FPS | - | 120 | 120 | 120 | 120 | 120 | 120 | 120 |
| Length (sec) | - | 5 | 5 | 5 | 5 | 5 | 2.5 | 5 |
| Num. of Frames | 132 | 600 | 600 | 600 | 600 | 600 | 300 | 600 |
| Resolutions | $720 \times 1080$ | $1920 \times 1080$ | $1920 \times 1080$ | $1920 \times 1080$ | $1920 \times 1080$ | $1920 \times 1080$ | $1920 \times 1080$ | $1920 \times 1080$ |

at 50 FPS and 12 seconds. The full names of each video in the UVG dataset are as follows: *2.city : CityAlley, 4.focus : FlowerFocus, 6.kids : FlowerKids, 8.pan : FlowerPan, 10.lips : Lips, 12.race : RaceNight, 14.river : RiverBank, 16.sunbath : SunBath, 17.twilight : Twilight.*

Table 6: the UVG17 Video Sessions.

| | Video Sessions | | | | | | | | |
|---|---|---|---|---|---|---|---|---|---|
| | 1 | 2 | 3 | 4 | 5 | 6 | 7 | 8 | 9 |
| Categories | bunny | city | beauty | focus | bosphorus | kids | bee | pan | jockey |
| FPS | - | 50 | 120 | 50 | 120 | 50 | 120 | 50 | 120 |
| Length (sec) | - | 12 | 5 | 12 | 5 | 12 | 5 | 12 | 5 |
| Num. of Frames | 132 | 600 | 600 | 600 | 600 | 600 | 600 | 600 | 600 |
| Resolutions | $720 \times 1080$ | $1920 \times 1080$ | $1920 \times 1080$ | $1920 \times 1080$ | $1920 \times 1080$ | $1920 \times 1080$ | $1920 \times 1080$ | $1920 \times 1080$ | $1920 \times 1080$ |

| | Video Sessions | | | | | | | |
|---|---|---|---|---|---|---|---|---|
| | 10 | 11 | 12 | 13 | 14 | 15 | 16 | 17 |
| Categories | lips | setgo | race | shake | river | yacht | sunbath | twilight |
| FPS | 120 | 120 | 50 | 120 | 50 | 120 | 50 | 50 |
| Length (sec) | 5 | 5 | 12 | 2.5 | 12 | 5 | 6 | 12 |
| Num. of Frames | 600 | 600 | 600 | 300 | 600 | 600 | 300 | 600 |
| Resolutions | $1920 \times 1080$ | $1920 \times 1080$ | $1920 \times 1080$ | $1920 \times 1080$ | $1920 \times 1080$ | $1920 \times 1080$ | $1920 \times 1080$ | $1920 \times 1080$ |

*3) DAVIS (Densely Annotated VIdeo Segmentation) of 50 Video Sessions*: We prepare a large-scale sequential video dataset, the Densely Annotation Video Segmentation dataset (DAVIS) (Perazzi et al., 2016). To validate our algorithm and investigate the limitations, we conducted the experiments on 50 video sequences with 3455 frames with a high-quality, high-resolution (1080p).

### A.3 IMPREMENTATION DETAILS

*1) 7 Videos in UVG of 8 Video Sessions*: To utilize the same video frame with NeRV, we downloaded 7 videos from the UVG dataset and employed the following commands to extract frames from the YUV Y4M videos.

» Download file : **[title] 3840x2160 8bit YUV Y4M**

» Command : **ffmpeg -i [file_name] [path]/f%05d.png**

*2) 9 Videos in UVG of 17 Video Sessions*: To expand our usage of videos, we acquired an additional 9 videos from the UVG dataset that are exclusively available as YUV RAW videos with a resolution of 3840x2160. We then extracted and resized the frames using the following command.

» Download file : **[title] 3840x2160 10bit YUV RAW**

» Command : **ffmpeg -s 3840x2160 -pix_fmt yuv420p10le -i [file_name] -vf scale=1920:1080 -pix_fmt rgb24 [path]/f%05d.png**

Digiturk provides the video contents of the UVG dataset. The dataset videos are available online at https://ultravideo.fi/#main.

**EWC** (Kirkpatrick et al., 2017). We trained EWC on the two novel benchmark datasets as a regularized baseline. When training with a new video, the EWC penalty was adopted as a regularization term to alleviate catastrophic forgetting. The importance of the parameter was calculated through the diagonal component of the Fisher Information matrix, and the EWC penalty increased as the difference in the vital parameter increased as follows:

$$L_E(v_t^s) = L(v_t^s) + \frac{\lambda}{2} \sum_i F_i(\theta_i - \theta_{p,i}^*)^2, \tag{11}$$

where $L_E$ is the total loss for EWC learning, $F$ represents the Fisher information matrix, $\lambda$ is a hyperparameter to determine the importance of the previous video, $i$ denotes each parameter, and $\theta_p^*$

Table 7: PFNR's architecture of Fourier Subneural Operator (FSO), $f$-NeRV$*$. Note that PE denotes positional encoding.

| layer | Module | Upscale Factor | Output Size $(C \times W \times H)$ | Number of parameters |
|---|---|---|---|---|
| 0 | PE (w frame index) | - | $80 \times 1 \times 1$ | - |
|   | PE (w video index) | - | $80 \times 1 \times 1$ | - |
| 1 | STEM of fc1 | - | $512 \times 1 \times 1$ | 81,920 |
|   | STEM of fc2 | - | $112 \times 16 \times 9$ | 8,257,536 |
| 2 | NeRV2 block of conv | $5\times$ | $112 \times 80 \times 45$ | 2,825,200 |
|   | $f$-NeRV2 | $1\times$ | $112 \times 80 \times 45$ | 1,605,632 |
| 3 | NeRV3 block of conv | $2\times$ | $96 \times 160 \times 90$ | 387,456 |
|   | $f$-NeRV3 | $1\times$ | $96 \times 160 \times 90$ | 37,847,040 |
| 4 | NeRV4 block of conv | $2\times$ | $96 \times 320 \times 180$ | 332,160 |
| 5 | NeRV5 block of conv | $2\times$ | $96 \times 640 \times 360$ | 332,160 |
| 6 | NeRV6 block of conv | $2\times$ | $96 \times 1280 \times 720$ | 332,160 |
| 7 | Multi-head layer for a video session | - | $3 \times 1280 \times 720$ | 291 |

denotes the parameter after training with the previous video. We randomly sampled 10 frames per video to compute the Fisher diagonal and stored them in a replay buffer. The hyperparameter $\lambda$ was experimentally set to 2e6 to scale the EWC penalty.

**iCaRL** (Rebuffi et al., 2017). We trained iCaRL as a rehearsal-based baseline on the two novel benchmark datasets. To replay previous videos, we store a total of $m = 800$ frames in the replay buffer as an exemplar set, and as the training progresses, we save $m/s$ frames per video. The exemplar management method is similar to that in iCaRL, where we compute the average feature map within the video and select $m/s$ video frames that approximate the average feature map. For knowledge distillation, we randomly sample frames from the exemplar set of previous videos at each learning step and performed training with the current video, as follows:

$$L_C(v_t^s) = L(v_t^s) + \lambda \frac{1}{t-1} \sum_i^{t-1} L(v_i^*), \tag{12}$$

where $v_i^*$ is a frame sampled from example set of $i$th video, and $\lambda$ is a hyperparameter experimentally set to 0.5.

**ESMER** (Sarfraz et al., 2023). We trained ESMER on the two novel benchmark datasets as a current strong rehearsal-based baseline. Error-Sensitive Reservoir Sampling (ESMER) maintains episodic memory, which leverages the error history to pre-select low-loss samples as candidates for the buffer of 800 samples. At this time, we didn't use any noisy labels when training ESMER. We observed that the ESMER could not reduce forgetting in representations. It seems better suited for retaining information in image classification tasks rather than neural implicit representations. Lastly, compared with iCaRL, ESMER replies buffer at each iteration, leading to ineffective training cost and performance as shown in Table 9.

Table 8: Statistics of FPS and PSNR on the UVG17 Video Sessions

| Method | FPS / Resolution | Avg. PSNR | FPS / Resolution | Avg. PSNR |
|---|---|---|---|---|
| STL, NeRV Chen et al. (2021a)$^*$ | 120 / 1920 × 1080 | **35.97** | 50 / 1920 × 1080 | 35.56 |
| iCaRL Rebuffi et al. (2017)$^*$ | 120 / 1920 × 1080 | **21.94** | 50 / 1920 × 1080 | 21.06 |
| ESMER Sarfraz et al. (2023)$^*$ | 120 / 1920 × 1080 | **21.43** | 50 / 1920 × 1080 | 19.25 |
| WSN$^*$ | 120 / 1920 × 1080 | **27.01** | 50 / 1920 × 1080 | 25.95 |
| PFNR, $f$-NeRV2 | 120 / 1920 × 1080 | **29.65** | 50 / 1920 × 1080 | 28.36 |
| PFNR, $f$-NeRV3 | 120 / 1920 × 1080 | **31.53** | 50 / 1920 × 1080 | 31.47 |
| MTL (upper-bound) | 120 / 1920 × 1080 | **29.64** | 50 / 1920 × 1080 | 28.83 |

## A.4 ARCHITECTURE

In this paper, we set NeRV as the baseline architecture, and we present a detailed overview of our architecture, as shown in Table 7. Our architecture has two differences compared to the previous

NeRV. Firstly, to incorporate the outputs from the positional encoder of both the video index and frame index, we expand the input size of the first layer in the MLP from 80 to 160. Secondly, to enable Multi-Task Learning (MTL), we employ distinct head layers (multi-heads) for each video after the NeRV block. Apart from these modifications, our architecture remains consistent with the previous NeRV architecture. We stack five NeRV blocks with upscaling factors of 5, 2, 2, 2, and 2, respectively. The lowest channel width for output feature maps in the NeRV block is also set to 96. Comparing our architecture to the baseline NeRV, the number of parameters increases by 0.3433% to 12,567,560 for eight videos. Similarly, for seventeen videos, the number of parameters increases by 0.3642% to 12,570,179.

## A.5 Additional Results

**Dataset statistic of FPS and PSNR.** As shown in Table 5 and Table 6, we have extracted sample frames according to each video's FPS and Length (sec) during training. There are two kinds of videos: 300 and 600 frames. We train each video for 150 epochs. Statistically, the average PSNR of 120 FPS was better than those of 50 FPS, as shown Table 8. When considering 30 PSNR is known as a sufficient resolution, 50 FPS-based sequence training could be adequate using the proposed PFNR. Moreover, the number of frames does not seem to be a critical factor to PSNR when considering that the PSNR of video session 1 (bunny), with 132 ($720 \times 1080$) frames, is greater than the 30 PSNR score.

**PFNR's Structure.** We investigate the most expressive, effective, and efficient structure with Fourier Subneural Operator (FSO, Equation 4) for progressive neural implicit representations. To do so, we prepare the layer-wise FSO to maintain the output size of the baseline layer as shown in Table 7: the number of parameters of a spectral layer depends on input and output size, so as the layer increases, the parameters also increase. Table 9 shows the effectiveness of PFNR with spectral layers in terms of PSNR, BWT, and CAP. To acquire neural implicit representations, PFNR explores more diverse parameters than WSN regarding the same capacity.

Table 9: PSNR results with Fourier Subnueral Operator (FSO) layer ($f$-NeRV∗) (NeRV block, Table 7) on UVG8 Video Sessions with average PSNR and Backward Transfer (BWT), and Capacity (CAP). Note that ∗ denotes our reproduced results.

| Method | Video Sessions | | | | | | | | Avg. PSNR / BWT | CAP |
|---|---|---|---|---|---|---|---|---|---|---|
| | 1 | 2 | 3 | 4 | 5 | 6 | 7 | 8 | | |
| STL, NeRV Chen et al. (2021a)∗ | 39.66 | 36.28 | 38.14 | 42.03 | 36.58 | 29.22 | 37.27 | 31.45 | 36.33 / - | 800.00 % |
| EWC Kirkpatrick et al. (2017)∗ | 10.19 | 11.15 | 14.47 | 8.39 | 12.21 | 10.27 | 9.97 | 23.98 | 12.58 / -17.59 | 100.00 % |
| iCaRL Rebuffi et al. (2017)∗ | 30.84 | 26.30 | 27.28 | 34.48 | 20.90 | 17.28 | 30.33 | 24.64 | 26.51 / -3.90 | 100.00 % |
| ESMER Sarfraz et al. (2023)∗ | 31.71 | 23.09 | 24.15 | 28.03 | 17.30 | 13.81 | 12.45 | 24.57 | 21.92 / -9.99 | 100.00 % |
| WSN∗, c = 10.0 % | 27.81 | 30.66 | 29.30 | 33.06 | 22.16 | 18.40 | 27.81 | 22.97 | 26.52 / 0.0 | 28.00 % |
| WSN∗, c = 30.0 % | 31.37 | 32.19 | 29.92 | 33.62 | 22.82 | 18.96 | 28.43 | 23.40 | 27.59 / 0.0 | 59.00 % |
| WSN∗, c = 50.0 % | 34.05 | 32.28 | 29.98 | 32.88 | 22.15 | 18.61 | 27.68 | 23.64 | 27.66 / 0.0 | 77.00 % |
| WSN∗, c = 70.0 % | 35.62 | 32.08 | 29.46 | 31.37 | 21.60 | 18.13 | 27.33 | 22.61 | 27.28 / 0.0 | 91.00 % |
| PFNR, c = 10.0 %, $f$-NeRV2 | 28.49 | 32.30 | 30.30 | 35.12 | 24.10 | 19.82 | 29.89 | 24.76 | 28.10 / 0.0 | 33.83 % |
| PFNR, c = 30.0 %, $f$-NeRV2 | 31.99 | 33.56 | 31.82 | 36.61 | 25.28 | 20.97 | 31.07 | 25.73 | 29.63 / 0.0 | 76.76 % |
| PFNR, c = 50.0 %, $f$-NeRV2 | 34.46 | 33.91 | 32.17 | 36.43 | 25.26 | 20.74 | 30.18 | 25.45 | 29.82 / 0.0 | 102.64 % |
| PFNR, c = 70.0 %, $f$-NeRV2 | 36.04 | 33.46 | 31.05 | 32.57 | 23.40 | 19.41 | 28.31 | 24.31 | 28.57 / 0.0 | 111.66 % |
| MTL (upper-bound) | 34.22 | 32.79 | 32.34 | 38.33 | 25.30 | 22.44 | 33.73 | 27.05 | 30.78 / - | 100.00 % |

**Large-scale Sequence.** We conducted the large-scale video sequence training to show its effectiveness and investigate the limitation of architecture parameters as shown in Table 10. The overall PFNR performances (STL, WSN, PFNR, and MTL) on the DAVIS50 dataset on the UVG8/17 dataset are lower than those on the UVG17 dataset. However, the performance trends of PFNR observed on the DAVIS50 dataset are consistent with the experimental results obtained from the UVG8/17 dataset. Regarding sequential and multi-task learning, the current performance falls short of achieving the target of 30 PSNR. This indicates a need for future work focused on designing model structures capable of continual learning on large-scale datasets.

**Forget-free Transfer Matrix.** We prepare the transfer matrix to prove our PFNR's forget-freeness and to show video correlation among other videos, as shown in Figure 6 on the UVG17 dataset; lower triangular estimated by each session subnetwork denotes that our PFNR is a forget-free method and upper triangular calculated by current session subnetwork denotes the video similarity between source and target. The PFNR proves the effectiveness from the lower triangular of Figure 6 (a) and

Table 10: PSNR results with Fourier Subnueral Operator (FSO) layer ($f$-NeRV∗) (NeRV block, Table 7) on DAVIS50 Video Sessions with average PSNR and Backward Transfer (BWT), and Capacity (CAP). Note that ∗ denotes our reproduced results.

| Method | Video Sessions | | | | | Avg. PSNR / BWT |
| --- | --- | --- | --- | --- | --- | --- |
| | **10** | **20** | **30** | **40** | **50** | |
| STL, NeRV Chen et al. (2021a)∗ | 28.92 | 31.10 | 34.96 | 29.35 | 28.87 | 31.41 / - |
| EWC Kirkpatrick et al. (2017)∗ | 10.57 | 12.87 | 15.22 | 11.25 | 10.45 | 12.24 / -18.23 |
| iCaRL Rebuffi et al. (2017)∗ | 15.22 | 18.55 | 17.59 | 13.47 | 15.89 | 16.54 / -8.48 |
| ESMER Sarfraz et al. (2023)∗ | 13.54 | 15.46 | 16.78 | 12.48 | 13.22 | 14.78 / -15.95 |
| WSN∗, c = 30.0 % | 19.20 | 20.80 | 23.39 | 21.56 | 21.45 | 21.56 / 0.0 |
| PFNR, c = 30.0 %, $f$-NeRV2 | 23.14 | 23.14 | 23.14 | 23.14 | 23.08 | 24.22 / 0.0 |
| PFNR, c = 30.0 %, $f$-NeRV3 | **25.58** | **27.79** | **31.42** | **27.22** | **24.88** | **27.57 / 0.0** |
| MTL (upper-bound) | 23.10 | 23.19 | 24.63 | 22.84 | 23.45 | 24.57 / - |

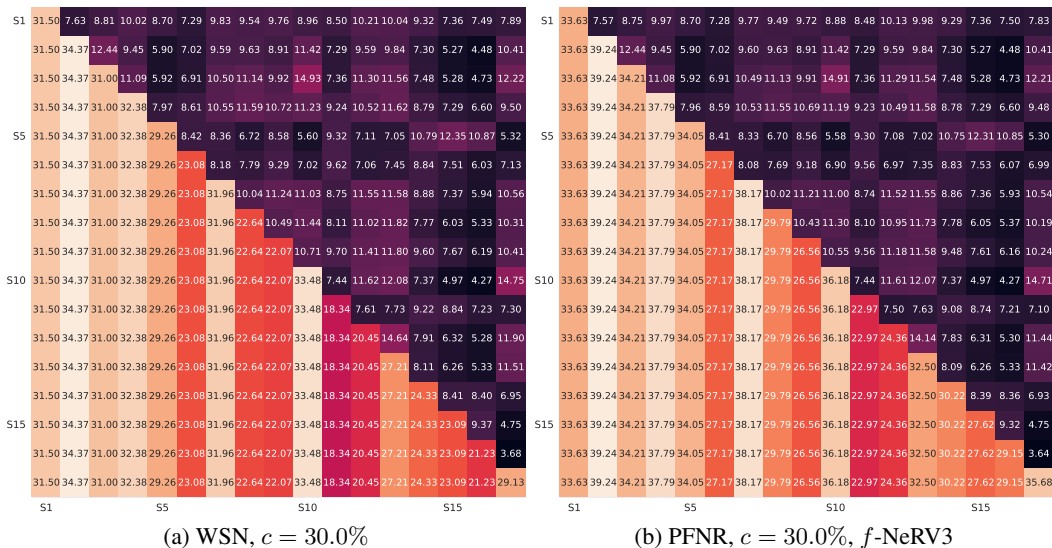

(a) WSN, $c = 30.0\%$          (b) PFNR, $c = 30.0\%$, $f$-NeRV3

Figure 6: Transfer Matrixes of WSN v.s. PFNR on the UVG17 dataset measured by PSNR of source and target.

(b). Nothing special is observable from the upper triangular since they are not correlated, however, there might be some shared representations.

**Ablation Study of FSO.** We prepare several ablation studies to prove the effectiveness of FSO. First, we show the performances of only real part (ignore an imaginary part) in f-NeRV2/3 as shown in Table 11. The PSNR performances of only real part were lower than those of both real and imaginary parts in f-NeRV2/3. We infer that the imaginary part of the winning ticket improves the implicit neural representations. Second, we also investigate the effectiveness of only FSO without Conv. Layer in f-NeRV2/3, as shown in Table 12. The PSNR performances were lower than FSO with Conv block. Therefore, the ensemble of FSO and Conv improves the implicit representations. Lastly, we investigate the effectiveness of sparse FSO in STL, as shown in Table 13. The sparse FSO boots the PSNR performances in STL. These ablation studies further strengthen the effectiveness of FSO for sequential neural implicit representations.

Table 11: PSNR results with Fourier Subnueral Operator (FSO) layer ($f$-NeRV∗) (detailed in Table 7) on UVG8 Video Sessions with average PSNR and Backward Transfer (BWT). Note that *w/o imag.* ignores the imaginary part in $f$-NeRV∗.

| Method | Video Sessions | | | | | | | | Avg. PSNR / BWT |
| --- | --- | --- | --- | --- | --- | --- | --- | --- | --- |
| | **1** | **2** | **3** | **4** | **5** | **6** | **7** | **8** | |
| PFNR, c = 50.0 %, $f$-NeRV2 | **34.46** | **33.91** | **32.17** | **36.43** | **25.26** | **20.74** | **30.18** | **25.45** | **29.82 / 0.0** |
| PFNR, c = 50.0 %, $f$-NeRV2 **w/o imag.** | 34.34 | 33.79 | 32.04 | 36.40 | 25.11 | 20.59 | 30.17 | 25.27 | 29.71 / 0.0 |
| PFNR, c = 50.0 %, $f$-NeRV3 | **36.45** | **35.15** | **35.10** | **38.57** | **28.07** | **23.06** | **32.83** | **27.70** | **32.12 / 0.0** |
| PFNR, c = 50.0 %, $f$-NeRV3 **w/o imag.** | 35.66 | 34.65 | 34.09 | 37.95 | 25.80 | 21.94 | 32.17 | 26.91 | 31.15 / 0.0 |

Table 12: PSNR results with Fourier Subnueral Operator (FSO) layer ($f$-NeRV∗) (detailed in Table 7) on UVG8 Video Sessions with average PSNR and Backward Transfer (BWT). Note that *w/o conv.* ignores the conv. layer in $f$-NeRV∗.

| Method | Video Sessions | | | | | | | | Avg. PSNR / BWT |
|---|---|---|---|---|---|---|---|---|---|
| | 1 | 2 | 3 | 4 | 5 | 6 | 7 | 8 | |
| PFNR, c = 50.0 %, $f$-NeRV2 | **34.46** | **33.91** | **32.17** | **36.43** | **25.26** | **20.74** | **30.18** | **25.45** | **29.82 / 0.0** |
| PFNR, c = 50.0 %, $f$-NeRV2 **w/o conv.** | 30.05 | 32.10 | 30.12 | 31.82 | 24.00 | 19.60 | 28.21 | 24.47 | 27.54 / 0.0 |
| PFNR, c = 50.0 %, $f$-NeRV3 | **36.45** | **35.15** | **35.10** | **38.57** | **28.07** | **23.06** | **32.83** | **27.70** | **32.12 / 0.0** |
| PFNR, c = 50.0 %, $f$-NeRV3 **w/o conv.** | 35.46 | 35.06 | 34.98 | 38.23 | 28.00 | 22.98 | 32.57 | 27.45 | 31.84 / 0.0 |

Table 13: PSNR results of STL with Fourier Subnueral Operator (FSO) layer ($f$-NeRV∗) (detailed in Table 7) on UVG8 Video Sessions with average PSNR and Backward Transfer (BWT).

| Method | Video Sessions | | | | | | | | Avg. PSNR / BWT |
|---|---|---|---|---|---|---|---|---|---|
| | 1 | 2 | 3 | 4 | 5 | 6 | 7 | 8 | |
| STL, NeRV Chen et al. (2021a)* | 39.66 | 36.28 | 38.14 | 42.03 | 36.58 | 29.22 | 37.27 | 31.45 | 36.33 / - |
| STL, NeRV , $f$-NeRV2 | 39.73 | 36.30 | 38.29 | 42.03 | 36.64 | 29.25 | 37.35 | 31.65 | 36.40 / - |
| STL, NeRV , $f$-NeRV3 | **42.75** | **37.65** | **42.05** | **42.36** | **40.01** | **34.21** | **40.15** | **36.15** | **39.41 / -** |

**Training time and Decoding FPS.** We train and test two baselines (NeRV, ESMER) with $f$-NeRV2 using one GPU (TITAN V, 12G) with a single batch size to investigate the computational expenses and decoding FPS on the UVG8 dataset, as shown in Table 14. In STL, NeRV with $f$-NeRV2 costs more computational times than NeRV. In VCL, memory buffer-based methods, i.e., ESMER, cost more training time since they replay the memory buffer in sequential training. On the other hand, architecture-based methods, i.e., PFNR, provide parameter-efficient, faster, forget-free solutions in training while cost computation expenses in the decoding process. Considering these limitations and advantages, we would find a more parameter-efficient FSO algorithm in future work.

Table 14: Training time and Decoding FPS with Fourier Subnueral Operator (FSO) layer ($f$-NeRV2) (detailed in Table 7) on UVG8 Video Sessions

| Method | Training time [hours] | Decoding FPS |
|---|---|---|
| STL, NeRV Chen et al. (2021a)* | 13.10 | 56.88 |
| STL, NeRV, $f$-NeRV2 | 25.66 | 31.72 |
| ESMER Sarfraz et al. (2023)* | 101.71 | 56.93 |
| PFNR, $f$-NeRV2 | 31.66 | 31.55 |

**Video Generation.** We prepared some results of video generation as shown in Figure 7. At the human recognition level, approximately 30 PSNR provides a little bit of blurred generated images as shown in WSN's PSNR(29.26); if the PSNR score is greater than 30, we hardly distinguish the quality of sequential neural implicit representations (city session). Thus, our objective is to maintain the sequential neural implicit representation of the 30 PSNR score. We have achieved this target PSNR score on the UVG17 dataset. We also show the quantized and encoded PFNR's video generation results Figure 8. We demonstrate that a compressed sparse solution in FP8 (PFNR with $c = 30.0\%$, $f$-NeRV2) generates video representations sequentially without a significant quality drop. The results of the FP4 showed that the compressed model can not create image pixels in detail.

## A.6 LAYER-WISE REPRESENTATIONS

To investigate the property of FSO, we prepare the layer-wise representations, as shown in Figure 9. The representations of PFNR ($f$-NeRV3) focus on textures rather than objects at the NeRV3 layer in the video session of bosphorus. In the video session of bee, PFNR ($f$-NeRV3) also tends to capture local textures broadly at the NeRV3 layer. This behavior of $f$-NeRV3 leads to better generalization at final prediction than others, such as WSN and PFNR ($f$-NeRV2). Moreover, we conducted an ablation study to inspect the best sparsity of $f$-NeRV3, as shown in Figure 10. Specifically, the performances of PFNR, c=50.0% depend on the sparsity of $f$-NeRV3. We observed that $f$-NeRV3 with c=50.0% was the best sparsity.

## A.7 CURRENT LIMITATIONS AND FUTURE WORK

Since the parameters of FSO depend on the input/output feature map size, in this task, the deeper the FSO layer, the larger the parameters increase. Nevertheless, we found the most parameter-efficient

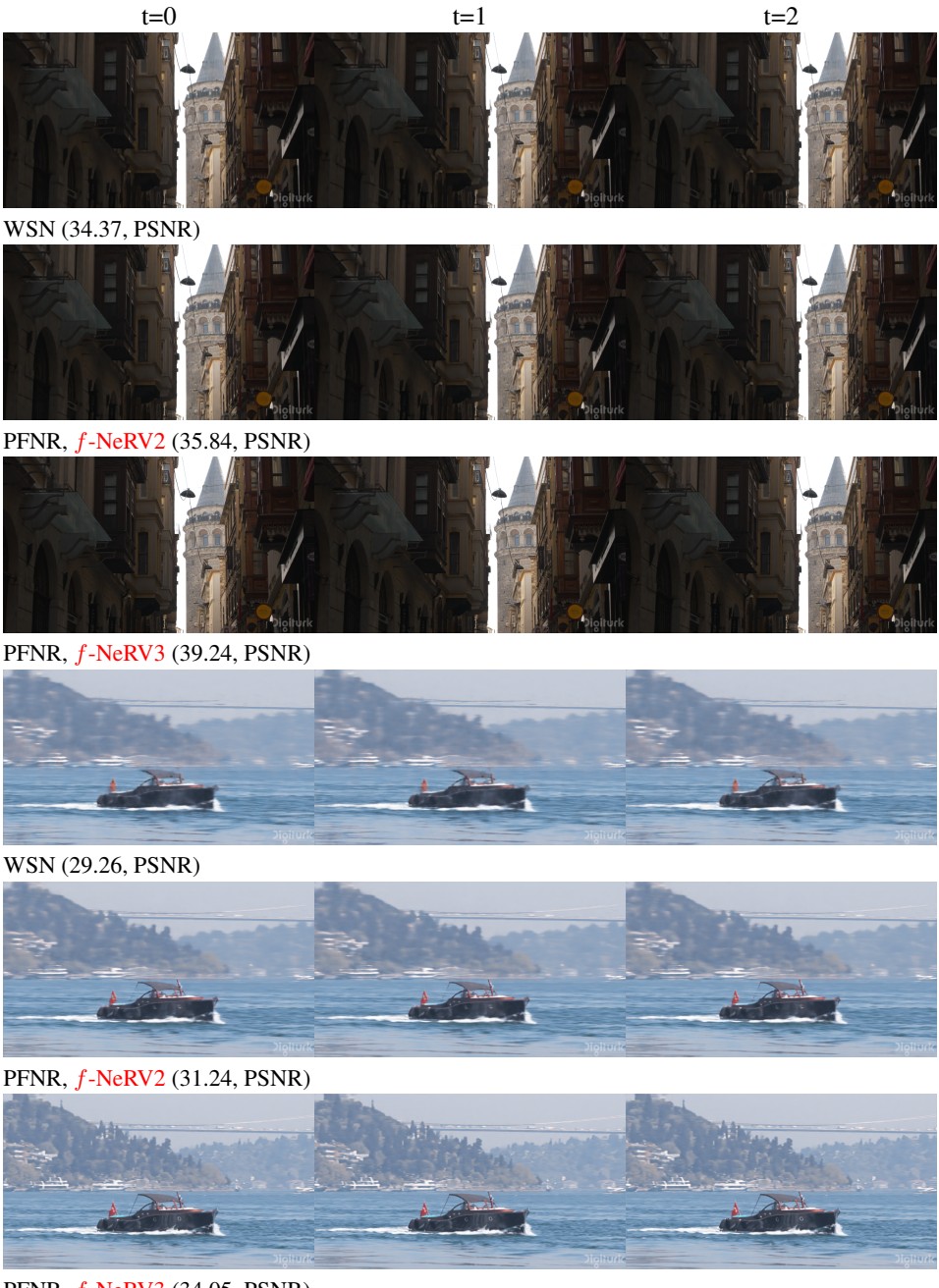

t=0    t=1    t=2

WSN (34.37, PSNR)

PFNR, $f$-NeRV2 (35.84, PSNR)

PFNR, $f$-NeRV3 (39.24, PSNR)

WSN (29.26, PSNR)

PFNR, $f$-NeRV2 (31.24, PSNR)

PFNR, $f$-NeRV3 (34.05, PSNR)

Figure 7: PFNR's Video Generation (from t=0 to t=2) with $c = 30.0\%$ on the UVG17 dataset.

FSO structure through layer-wise inspection and layer sparsity to describe the best neural implicit representations. In future work, we will design a more parameter-efficient FSO layer in continual tasks such as neural implicit representation and task/class incremental learnings.

### A.8 BROADER IMPACTS

As the most popular media format nowadays, videos are generally viewed as frames of sequences. Unlike that, our proposed PFNR is a novel way to represent sequential videos as a function of video session and time, parameterized by the neural network firstly in Fourier space, which is more efficient and might be used in many video-related tasks, such as sequential video compression, sequential

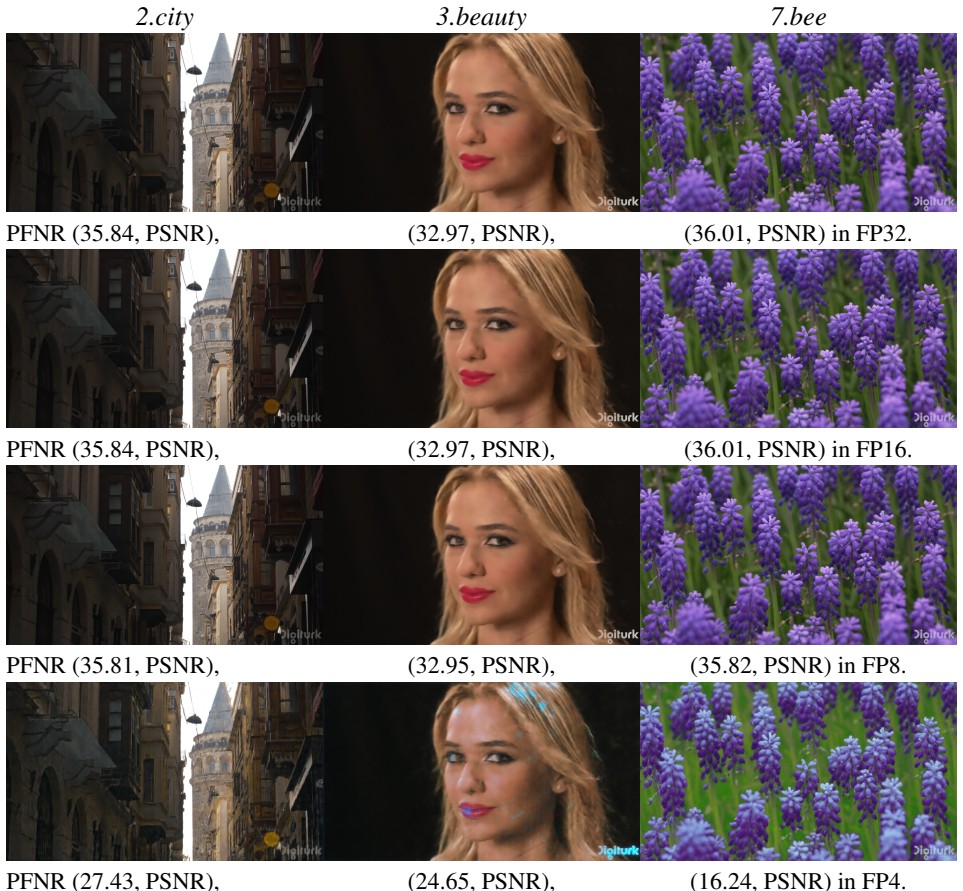

Figure 8: PFNR's Qunatizationed and Compresssed Video Generation (t=0) with $c = 30.0\%$, $f$-NeRV2 on the UVG17 dataset. Note that PFNR's prediction in FP32, 16, 8, and 4.

Table 15: PSNR results with Fourier Subnueral Operator (FSO) layer ($f$-NeRV∗) (detailed in Table 7) on UVG17 Video Sessions with average PSNR, Backward Transfer (BWT) of PSNR, Model Capacity (CAP). Note that ∗ denotes our reproduced results.

| Method | 1 | 2 | 3 | 4 | 5 | 6 | 7 | 8 | 9 | 10 | 11 | 12 | 13 | 14 | 15 | 16 | 17 | Avg. PSNR BWT | CAP |
|---|---|---|---|---|---|---|---|---|---|---|---|---|---|---|---|---|---|---|---|
| STL, NeRV Chen et al. (2021a)∗ | 39.66 | 44.89 | 36.28 | 41.13 | 38.14 | 31.53 | 42.03 | 34.74 | 36.58 | 36.85 | 29.22 | 31.81 | 37.27 | 34.18 | 31.45 | 38.41 | 43.86 | 36.94 / - | 1700.00 % |
| EWC Kirkpatrick et al. (2017)∗ | 11.15 | 9.21 | 12.71 | 11.40 | 15.58 | 9.25 | 7.06 | 12.96 | 6.34 | 10.31 | 9.55 | 13.39 | 5.76 | 8.67 | 10.93 | 10.92 | 28.29 | 11.38 / -16.13 | 100.00 % |
| iCaRL Rebuffi et al. (2017)∗ | 24.31 | 28.25 | 22.19 | 22.74 | 22.84 | 16.55 | 29.37 | 17.92 | 16.65 | 27.43 | 13.64 | 16.42 | 24.02 | 21.60 | 19.40 | 18.60 | 26.46 | 21.67 / -6.23 | 100.00 % |
| ESMER Sarfraz et al. (2023)∗ | 30.77 | 26.33 | 22.79 | 21.35 | 23.76 | 13.64 | 28.25 | 15.22 | 16.71 | 23.78 | 13.35 | 15.23 | 18.21 | 19.22 | 24.59 | 20.61 | 22.42 | 20.95 / -15.23 | 100.00 % |
| WSN∗, c = 10.0 % | 27.68 | 31.31 | 30.29 | 31.63 | 28.66 | 22.57 | 31.62 | 22.04 | 21.05 | 32.71 | 17.85 | 20.09 | 27.07 | 23.84 | 22.98 | 20.50 | 28.56 | 25.91 / 0.0 | 53.25 % |
| WSN∗, c = 30.0 % | 31.50 | 34.37 | 31.00 | 32.38 | 29.26 | 23.08 | 31.96 | 22.64 | 22.07 | 33.48 | 18.34 | 20.45 | 27.21 | 24.33 | 23.09 | 21.23 | 29.13 | 26.80 / 0.0 | 91.10 % |
| WSN∗, c = 50.0 % | 34.02 | 34.93 | 31.04 | 31.74 | 28.95 | 23.07 | 31.26 | 22.32 | 21.93 | 33.35 | 18.22 | 20.34 | 26.88 | 24.22 | 22.72 | 21.30 | 28.86 | 26.77 / 0.0 | 97.23 % |
| WSN∗, c = 70.0 % | 35.64 | 34.36 | 30.26 | 30.27 | 27.99 | 22.55 | 29.88 | 21.46 | 20.79 | 32.37 | 17.63 | 20.00 | 26.68 | 23.79 | 22.34 | 20.69 | 28.68 | 26.20 / 0.0 | 99.01 % |
| PFNR, c = 10.0 %, $f$-NeRV2 | 28.31 | 33.57 | 31.92 | 33.67 | 29.98 | 23.99 | 34.39 | 24.8 | 23.94 | 35.08 | 19.70 | 22.03 | 29.56 | 26.57 | 24.79 | 24.10 | 31.35 | 28.10 / 0.0 | 59.58 % |
| PFNR, c = 30.0 %, $f$-NeRV2 | 32.01 | 35.84 | 32.97 | 35.17 | 31.24 | 24.82 | 36.01 | 25.85 | 24.83 | 35.76 | 20.50 | 22.79 | 30.40 | 27.37 | 25.52 | 25.40 | 32.70 | 29.36 / 0.0 | 100.01 % |
| PFNR, c = 50.0 %, $f$-NeRV2 | 34.49 | 37.13 | 33.21 | 35.50 | 30.87 | 24.72 | 34.36 | 24.79 | 24.73 | 35.65 | 20.33 | 22.65 | 29.78 | 27.05 | 25.18 | 25.18 | 32.39 | 29.29 / 0.0 | 105.33 % |
| PFNR, c = 70.0 %, $f$-NeRV2 | 36.02 | 36.50 | 32.09 | 32.15 | 28.67 | 23.35 | 30.63 | 22.86 | 23.18 | 34.90 | 19.08 | 21.30 | 27.87 | 25.86 | 24.12 | 23.47 | 30.34 | 27.79 / 0.0 | 112.33 % |
| MTL (upper-bound) | 32.39 | 34.35 | 31.45 | 34.03 | 30.70 | 24.53 | 37.13 | 27.83 | 23.80 | 34.69 | 20.77 | 22.37 | 32.71 | 28.00 | 25.89 | 26.40 | 33.16 | 29.42 / - | 100.00 % |

video denoising, complex sequential physical modeling (Li et al., 2020a;b; Kovachki et al., 2021; Tran et al., 2021), and so on. Hopefully, this can save bandwidth and fasten media streaming, enriching entertainment potential. Unfortunately, like many advances in deep learning for videos, this approach could be used for various purposes beyond our control.

A.9 ACKNOWLEDGEMENT

This work was partly supported by Institute for Information communications Technology Planning Evaluation (IITP) grant funded by the Korea government(MSIT) (No. 2021-0-01381, Development

Table 16: MS-SSIM results with Fourier Subnueral Operator (FSO) layer ($f$-NeRV∗) (detailed in Table 7) on UVG17 Video Sessions with average MS-SSIM and Backward Transfer (BWT) of MS-SSIM. Note that ∗ denotes our reproduced results.

| Method | \multicolumn{17}{c}{Video Sessions} | Avg. MS-SSIM BWT |
|---|---|---|---|---|---|---|---|---|---|---|---|---|---|---|---|---|---|---|
| | 1 | 2 | 3 | 4 | 5 | 6 | 7 | 8 | 9 | 10 | 11 | 12 | 13 | 14 | 15 | 16 | 17 | |
| STL, NeRV* Chen et al. (2021a) | 0.99 | 0.99 | 0.95 | 0.98 | 0.98 | 0.96 | 0.99 | 0.98 | 0.97 | 0.95 | 0.96 | 0.96 | 0.98 | 0.98 | 0.96 | 0.99 | 0.99 | 0.97 / - |
| EWC Kirkpatrick et al. (2017)* | 0.26 | 0.24 | 0.44 | 0.24 | 0.40 | 0.29 | 0.15 | 0.17 | 0.26 | 0.26 | 0.17 | 0.34 | 0.04 | 0.30 | 0.33 | 0.31 | 0.91 | 0.30 / -0.55 |
| iCaRL Rebuffi et al. (2017)* | 0.74 | 0.88 | 0.67 | 0.67 | 0.64 | 0.48 | 0.91 | 0.53 | 0.37 | 0.82 | 0.35 | 0.53 | 0.75 | 0.70 | 0.61 | 0.60 | 0.87 | 0.65 / -0.20 |
| ESMER Sarfraz et al. (2023)* | 0.85 | 0.86 | 0.64 | 0.63 | 0.66 | 0.46 | 0.89 | 0.51 | 0.42 | 0.79 | 0.30 | 0.51 | 0.43 | 0.68 | 0.82 | 0.6 | 0.63 | 0.62 / -0.37 |
| WSN*, c = 10.0 % | 0.90 | 0.94 | 0.88 | 0.92 | 0.87 | 0.75 | 0.96 | 0.74 | 0.69 | 0.91 | 0.57 | 0.72 | 0.86 | 0.80 | 0.74 | 0.69 | 0.92 | 0.82 / 0.0 |
| WSN*, c = 30.0 % | 0.96 | 0.97 | 0.89 | 0.93 | 0.88 | 0.77 | 0.97 | 0.77 | 0.73 | 0.91 | 0.60 | 0.74 | 0.86 | 0.81 | 0.76 | 0.72 | 0.93 | 0.84 / 0.0 |
| WSN*, c = 50.0 % | 0.98 | 0.97 | 0.89 | 0.92 | 0.88 | 0.77 | 0.96 | 0.75 | 0.73 | 0.91 | 0.60 | 0.74 | 0.85 | 0.80 | 0.75 | 0.73 | 0.92 | 0.83 / 0.0 |
| WSN*, c = 70.0 % | 0.98 | 0.97 | 0.88 | 0.91 | 0.85 | 0.75 | 0.95 | 0.70 | 0.68 | 0.91 | 0.55 | 0.72 | 0.85 | 0.80 | 0.73 | 0.70 | 0.92 | 0.82 / 0.0 |
| PFNR, c = 10.0 %, $f$-NeRV2 | 0.92 | 0.97 | 0.90 | 0.94 | 0.90 | 0.81 | 0.98 | 0.86 | 0.79 | 0.93 | 0.69 | 0.79 | 0.91 | 0.87 | 0.82 | 0.84 | 0.96 | 0.88 / 0.0 |
| PFNR, c = 30.0 %, $f$-NeRV2 | 0.97 | 0.98 | 0.92 | 0.95 | 0.92 | 0.84 | 0.98 | 0.88 | 0.82 | 0.93 | 0.73 | 0.82 | 0.92 | 0.89 | 0.84 | 0.87 | 0.97 | 0.90 / 0.0 |
| PFNR, c = 50.0 %, $f$-NeRV2 | 0.98 | 0.99 | 0.92 | 0.95 | 0.92 | 0.83 | 0.98 | 0.86 | 0.81 | 0.93 | 0.72 | 0.82 | 0.91 | 0.88 | 0.84 | 0.87 | 0.97 | 0.89 / 0.0 |
| PFNR, c = 70.0 %, $f$-NeRV2 | 0.99 | 0.98 | 0.91 | 0.93 | 0.87 | 0.78 | 0.96 | 0.77 | 0.77 | 0.93 | 0.66 | 0.77 | 0.89 | 0.85 | 0.80 | 0.82 | 0.95 | 0.86 / 0.0 |
| PFNR, c = 10.0 %, $f$-NeRV3 | 0.96 | 0.99 | 0.92 | 0.96 | 0.94 | 0.86 | 0.99 | 0.94 | 0.82 | 0.93 | 0.76 | 0.83 | 0.93 | 0.92 | 0.87 | 0.90 | 0.98 | 0.91 / 0.0 |
| PFNR, c = 30.0 %, $f$-NeRV3 | 0.98 | 0.99 | 0.93 | 0.97 | 0.96 | 0.91 | 0.99 | 0.96 | 0.87 | 0.94 | 0.84 | 0.87 | 0.94 | 0.94 | 0.90 | 0.94 | 0.98 | 0.94 / 0.0 |
| PFNR, c = 50.0 %, $f$-NeRV3 | 0.99 | 0.99 | 0.93 | 0.97 | 0.96 | 0.91 | 0.99 | 0.95 | 0.87 | 0.94 | 0.83 | 0.88 | 0.94 | 0.94 | 0.9 | 0.95 | 0.98 | 0.94 / 0.0 |
| PFNR, c = 70.0 %, $f$-NeRV3 | 0.99 | 0.99 | 0.93 | 0.96 | 0.93 | 0.87 | 0.98 | 0.90 | 0.81 | 0.93 | 0.76 | 0.83 | 0.93 | 0.92 | 0.87 | 0.91 | 0.97 | 0.91 / 0.0 |
| MTL (upper-bound) | 0.97 | 0.97 | 0.90 | 0.94 | 0.91 | 0.82 | 0.99 | 0.92 | 0.80 | 0.92 | 0.75 | 0.81 | 0.94 | 0.90 | 0.85 | 0.89 | 0.97 | 0.90 / - |

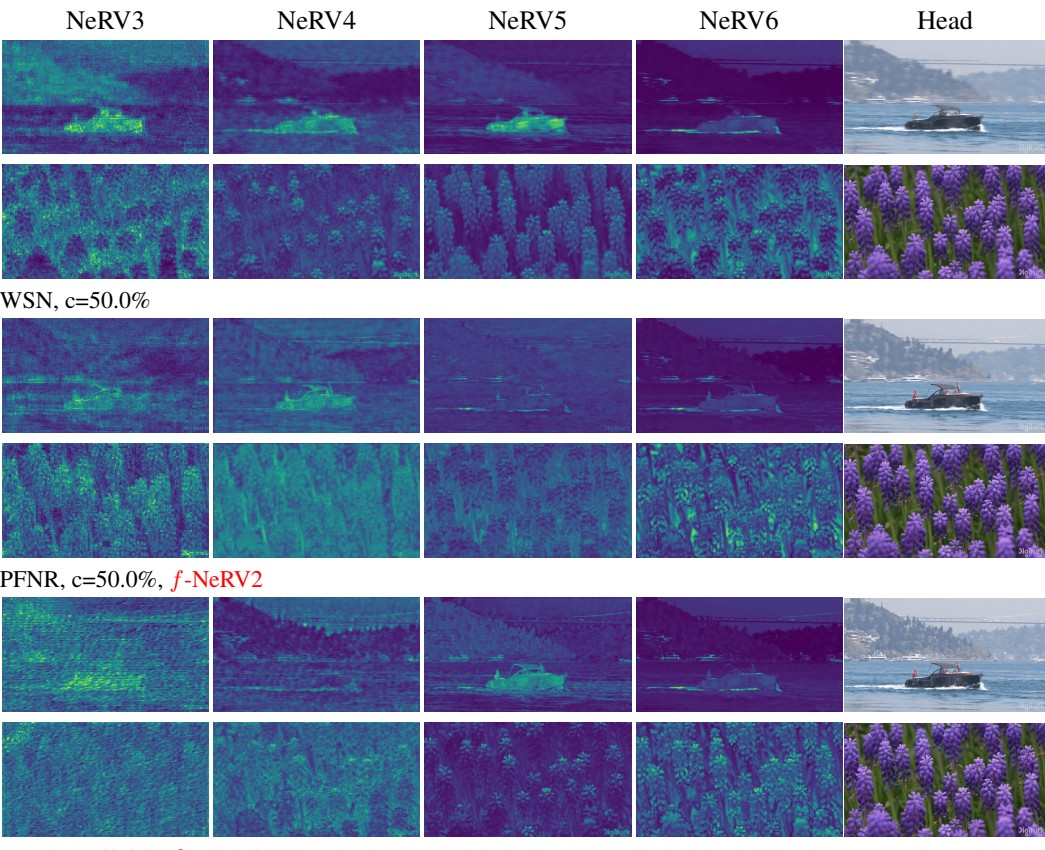

Figure 9: PFNR's Representations of NeRV Blocks with $c = 50.0\%$ on the UVG17 dataset.

of Causal AI through Video Understanding and Reinforcement Learning, and Its Applications to Real Environments) and partly supported by the National Research Foundation of Korea (NRF) grant funded by the Korea government(MSIT) (No. 2022R1A2C2012706)

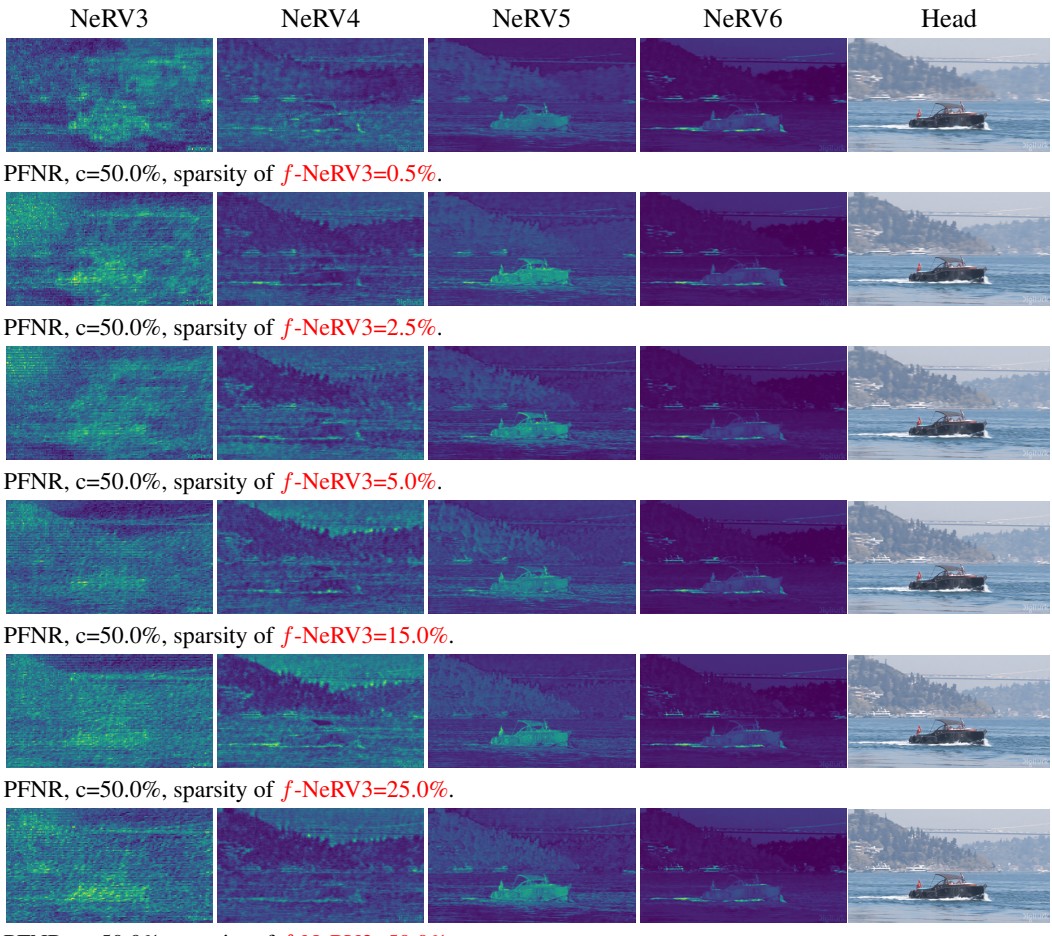

Figure 10: Various sparsity of $f$-NeRV3 ranging from 0.05 % (top row) to 50.0 % (bottom row) on the UVG17 dataset.

## A.10 PUBLIC CODES OF OUR PFNR

We provide the core parts of PFNR to understand better. Please refer to the attached file. We will provide all training and inference codes soon.

Table 17: PFNR ($f$-NeRV2) of Quantization + Compression on the UVG17 Video Sessions with average PSNR/MS-SSIM and Backward Transfer (BTW). Note that bit is the bit length used to represent parameter value.

| Method | Video Sessions | | | | | | | | | | | | | | | | | Avg. * / BWT |
|---|---|---|---|---|---|---|---|---|---|---|---|---|---|---|---|---|---|---|
| | 1 | 2 | 3 | 4 | 5 | 6 | 7 | 8 | 9 | 10 | 11 | 12 | 13 | 14 | 15 | 16 | 17 | |
| **c = 10.0%,** | PSNR | | | | | | | | | | | | | | | | | |
| bit=4 | 17.73 | 30.45 | 30.74 | 30.56 | 26.15 | 23.29 | 23.16 | 19.75 | 23.80 | 26.19 | 19.53 | 20.36 | 24.99 | 26.28 | 24.61 | 23.86 | 29.95 | 24.79 / -0.80 |
| bit=8 | 28.21 | 33.56 | 31.92 | 33.66 | 29.97 | 23.98 | 34.35 | 24.78 | 23.95 | 35.07 | 19.70 | 22.02 | 29.57 | 26.58 | 24.79 | 24.09 | 31.34 | 28.09 / -0.01 |
| bit=16 | 28.31 | 33.57 | 31.92 | 33.67 | 29.98 | 23.99 | 34.39 | 24.80 | 23.94 | 35.08 | 19.70 | 22.03 | 29.56 | 26.57 | 24.79 | 24.10 | 31.35 | 28.10 / 0.00 |
| bit=32 | 28.31 | 33.57 | 31.92 | 33.67 | 29.98 | 23.99 | 34.39 | 24.80 | 23.94 | 35.08 | 19.70 | 22.03 | 29.56 | 26.57 | 24.79 | 24.10 | 31.35 | 28.10 / 0.00 |
| | MS-SSIM | | | | | | | | | | | | | | | | | |
| bit=4 | 0.60 | 0.94 | 0.90 | 0.93 | 0.87 | 0.80 | 0.89 | 0.73 | 0.79 | 0.89 | 0.69 | 0.76 | 0.87 | 0.86 | 0.82 | 0.83 | 0.95 | 0.83 / -0.02 |
| bit=8 | 0.92 | 0.97 | 0.90 | 0.94 | 0.90 | 0.81 | 0.98 | 0.86 | 0.79 | 0.93 | 0.69 | 0.79 | 0.91 | 0.87 | 0.82 | 0.84 | 0.96 | 0.88 / 0.00 |
| bit=16 | 0.92 | 0.97 | 0.90 | 0.94 | 0.90 | 0.81 | 0.98 | 0.86 | 0.79 | 0.93 | 0.69 | 0.79 | 0.91 | 0.87 | 0.82 | 0.84 | 0.96 | 0.88 / 0.00 |
| bit=32 | 0.92 | 0.97 | 0.90 | 0.94 | 0.90 | 0.81 | 0.98 | 0.86 | 0.79 | 0.93 | 0.69 | 0.79 | 0.91 | 0.87 | 0.82 | 0.84 | 0.96 | 0.88 / 0.00 |
| **c = 30.0%,** | PSNR | | | | | | | | | | | | | | | | | |
| bit=4 | 10.96 | 27.43 | 24.65 | 25.04 | 24.05 | 22.92 | 16.24 | 23.08 | 23.91 | 33.51 | 20.10 | 22.47 | 28.79 | 26.58 | 24.94 | 24.87 | 31.61 | 24.19 / -2.13 |
| **bit=8** | **31.72** | **35.81** | **32.95** | **35.11** | **31.22** | **24.81** | **35.82** | **25.84** | **24.84** | **35.76** | **20.49** | **22.79** | **30.40** | **27.37** | **25.52** | **25.40** | **32.69** | **29.33 / -0.02** |
| bit=16 | 32.01 | 35.84 | 32.97 | 35.17 | 31.24 | 24.82 | 36.01 | 25.85 | 24.83 | 35.76 | 20.50 | 22.79 | 30.40 | 27.37 | 25.52 | 25.40 | 32.70 | 29.36 / 0.00 |
| bit=32 | 32.01 | 35.84 | 32.97 | 35.17 | 31.24 | 24.82 | 36.01 | 25.85 | 24.83 | 35.76 | 20.50 | 22.79 | 30.40 | 27.37 | 25.52 | 25.40 | 32.70 | 29.36 / 0.00 |
| | MS-SSIM | | | | | | | | | | | | | | | | | |
| bit=4 | 0.47 | 0.92 | 0.85 | 0.91 | 0.84 | 0.81 | 0.74 | 0.82 | 0.80 | 0.93 | 0.72 | 0.81 | 0.91 | 0.88 | 0.83 | 0.86 | 0.96 | 0.83 / -0.04 |
| bit=8 | 0.97 | 0.98 | 0.92 | 0.95 | 0.92 | 0.84 | 0.98 | 0.88 | 0.82 | 0.93 | 0.73 | 0.82 | 0.92 | 0.89 | 0.84 | 0.87 | 0.97 | 0.90 / 0.00 |
| bit=16 | 0.97 | 0.98 | 0.92 | 0.95 | 0.92 | 0.84 | 0.98 | 0.88 | 0.82 | 0.93 | 0.73 | 0.82 | 0.92 | 0.89 | 0.84 | 0.87 | 0.97 | 0.90 / 0.00 |
| bit=32 | 0.97 | 0.98 | 0.92 | 0.95 | 0.92 | 0.84 | 0.98 | 0.88 | 0.82 | 0.93 | 0.73 | 0.82 | 0.92 | 0.89 | 0.84 | 0.87 | 0.97 | 0.90 / 0.00 |
| **c = 50.0%,** | PSNR | | | | | | | | | | | | | | | | | |
| bit=4 | 7.19 | 23.92 | 20.83 | 24.71 | 24.77 | 21.91 | 28.55 | 23.03 | 23.33 | 32.96 | 19.22 | 21.80 | 25.86 | 22.00 | 23.85 | 22.82 | 29.59 | 22.92 / -3.91 |
| bit=8 | 34.03 | 37.08 | 33.20 | 35.47 | 30.86 | 24.71 | 34.34 | 24.78 | 24.73 | 35.64 | 20.33 | 22.65 | 29.78 | 27.04 | 25.17 | 25.17 | 32.39 | 29.26 / -0.03 |
| bit=16 | 34.49 | 37.13 | 33.21 | 35.50 | 30.87 | 24.72 | 34.36 | 24.79 | 24.73 | 35.65 | 20.33 | 22.65 | 29.78 | 27.05 | 25.18 | 25.18 | 32.39 | 29.29 / 0.00 |
| bit=32 | 34.49 | 37.13 | 33.21 | 35.50 | 30.87 | 24.72 | 34.36 | 24.79 | 24.73 | 35.65 | 20.33 | 22.65 | 29.78 | 27.05 | 25.18 | 25.18 | 32.39 | 29.29 / 0.00 |
| | MS-SSIM | | | | | | | | | | | | | | | | | |
| bit=4 | 0.25 | 0.87 | 0.79 | 0.87 | 0.85 | 0.78 | 0.95 | 0.81 | 0.79 | 0.92 | 0.68 | 0.78 | 0.88 | 0.80 | 0.81 | 0.82 | 0.94 | 0.80 / -0.07 |
| bit=8 | 0.98 | 0.99 | 0.92 | 0.95 | 0.92 | 0.83 | 0.98 | 0.86 | 0.81 | 0.93 | 0.72 | 0.82 | 0.91 | 0.88 | 0.84 | 0.87 | 0.97 | 0.89 / 0.00 |
| bit=16 | 0.98 | 0.99 | 0.92 | 0.95 | 0.92 | 0.83 | 0.98 | 0.86 | 0.81 | 0.93 | 0.72 | 0.82 | 0.91 | 0.88 | 0.84 | 0.87 | 0.97 | 0.89 / 0.00 |
| bit=32 | 0.98 | 0.99 | 0.92 | 0.95 | 0.92 | 0.83 | 0.98 | 0.86 | 0.81 | 0.93 | 0.72 | 0.82 | 0.91 | 0.88 | 0.84 | 0.87 | 0.97 | 0.89 / 0.00 |
| **c = 70.0%,** | PSNR | | | | | | | | | | | | | | | | | |
| bit=4 | 7.05 | 15.54 | 16.27 | 18.14 | 25.63 | 21.95 | 24.75 | 20.50 | 20.97 | 17.84 | 18.34 | 19.18 | 23.49 | 14.20 | 22.55 | 22.04 | 17.19 | 19.15/ -3.48 |
| bit=8 | 35.52 | 36.43 | 32.08 | 32.14 | 28.66 | 23.35 | 30.61 | 22.86 | 23.18 | 34.89 | 19.08 | 21.29 | 21.86 | 25.86 | 24.13 | 23.47 | 30.30 | 27.75 / -0.03 |
| bit=16 | 36.02 | 36.50 | 32.09 | 32.15 | 28.67 | 23.35 | 30.63 | 22.86 | 23.18 | 34.90 | 19.08 | 21.30 | 27.87 | 25.86 | 24.12 | 23.47 | 30.34 | 27.79 / 0.00 |
| bit=32 | 36.02 | 36.50 | 32.09 | 32.15 | 28.67 | 23.35 | 30.63 | 22.86 | 23.18 | 34.90 | 19.08 | 21.30 | 27.87 | 25.86 | 24.12 | 23.47 | 30.34 | 27.79 / 0.00 |
| | MS-SSIM | | | | | | | | | | | | | | | | | |
| bit=4 | 0.28 | 0.56 | 0.68 | 0.76 | 0.82 | 0.74 | 0.88 | 0.69 | 0.71 | 0.75 | 0.61 | 0.71 | 0.79 | 0.55 | 0.75 | 0.78 | 0.88 | 0.70 / -0.08 |
| bit=8 | 0.98 | 0.98 | 0.91 | 0.93 | 0.87 | 0.78 | 0.96 | 0.77 | 0.77 | 0.93 | 0.65 | 0.77 | 0.89 | 0.85 | 0.80 | 0.82 | 0.95 | 0.86 / 0.00 |
| bit=16 | 0.99 | 0.98 | 0.91 | 0.93 | 0.87 | 0.78 | 0.96 | 0.77 | 0.77 | 0.93 | 0.66 | 0.77 | 0.89 | 0.85 | 0.80 | 0.82 | 0.95 | 0.86 / 0.00 |
| bit=32 | 0.99 | 0.98 | 0.91 | 0.93 | 0.87 | 0.78 | 0.96 | 0.77 | 0.77 | 0.93 | 0.66 | 0.77 | 0.89 | 0.85 | 0.80 | 0.82 | 0.95 | 0.86 / 0.00 |

