# OpenReview forum: "Progressive Fourier Neural Representation for Sequential Video Compilation"
_ICLR.cc/2024/Conference — ICLR 2024 poster_

### Official Review · Reviewer_H5Ez · 2023-11-01

**Soundness:** 2 fair
**Presentation:** 2 fair
**Contribution:** 2 fair
**Rating:** 6
**Confidence:** 3

**Summary:**

This paper addresses a novel practical task scenario where a pretrained video neural implicit representation needs to continually learn new data while keeping the learned information. It proposes a novel Progressive Fourier Neural Representation method to tackle this, which continuously learns a compact subnetwork for each video in Fourier space. The proposed method is tested on several datasets with multiple metrics including PSNR and SSIM, and proves to outperform the baseline and other competiters.

**Strengths:**

- The problem of encoding new videos into existing video INRs is important in practice, and the proposed method addresses it well according to the experiment results with higher PSNR and SSIM compared to other competitiers.

- Besides the detailedly listed final metric results, it also shows abundant ablations on several hyperparameters and such as which layer to choose for the proposed model to learn etc., in both the main paper and the supplementary.

- Many baselines are discussed in the related work section as well as compared technically in the experiments.

**Weaknesses:**

- Although the quantitative results are listed in details for every video in two settings, overall the proposed method is only tested on two datasets, and both UVG series. More diverse choices would make the results more solid, such as the DAVIS dataset series etc.

- The compression performance (model size) is not well tested and displayed, especially in the tables. Figure 2 and its paragraph discussed some but is still not clear to link with the values in other tables.

- The illustration of the proposed method is relatively limited, e.g. about the details in Fourier space, which might hide the novelty and complexity of the proposed method.

- Not many visual comparisons are provided especially in the main paper, and Figure 7 is not specifically explained on their differences and advantages etc.

- [Minor] Section 3.1 said that Figure 1 is one "possible" structure, while there isn't any other design mentioned in this paper. Maybe there can exist more variants but if not mentioned then Figure 1 is just exactly "our proposed structure" to be clearer.

- [Minor] Figure 1 is overall good but the font and diagram size is a bit small compared to frame images.

- [Minor] Sometimes the HNerv paper is noted as Nerv (but with the correct 2023 reference).

**Questions:**

- In the abstract and introduction, it is illustrated that since the INRs learn to memorize videos "regardless of data relevancy or similarity", it is hard for them to be generalized to multiple complex data and thus continual learning is important for the models to learn new videos without forgetting previously learned videos. I agree that memorizing videos regardless of data relevancy or similarity limits INRs' efficiency and scalability, but shouldn't this characteristic help an INR to learn multiple unrelated videos compared to those that learn with data relevancy or similarity?

---

> ### Author Response · Authors · 2023-11-16
> **Large-scale dataset (DAVIS50).**
>
> - **Large-scale dataset (DAVIS50)**
>
>     As reviewer MyBt and H5Ez suggested, we prepare a large-scale sequential video dataset, the Densely Annotation Video Segmentation dataset (DAVIS50). To validate our algorithm and investigate the limitations, we conducted the experiments on 50 video sequences with 3455 frames with a high-quality, high-resolution (1080p). We have included the primary results to show the effectiveness of our PFNR. For simplicity, the table includes the PSNR performances of 10, 20, 30, 40, and 50 video sessions.
>
>
>     | Method                     | 10    | 20    | 30    | 40    | 50    | Avg.PSNR/BWT |
>     |----------------------------|-------|-------|-------|-------|-------|--------------|
>     | STL,NeRV                   | 28.92 | 31.10 | 34.96 | 29.35 | 28.87 |  31.41 / -   |
>     | WSN, c=30.0 \%             | 19.20 | 20.80 | 23.39 | 21.56 | 21.45 |	21.56 / 0.0 |
>     | PFNR, c=30.0 \%, $f$-NeRV2 | 23.14 | 23.14 | 23.14 | 23.14 | 23.08 |	24.22 / 0.0 |
>     | PFNR, c=30.0 \%, $f$-NeRV3 | **25.58** | **27.79** | **31.42** | **27.22** | **24.88** |  **27.57** / **0.0** |
>     | MTL                        | 23.10 | 23.19 | 24.63 | 22.84 | 23.45 |  24.57 / -   |
>
>     The overall PFNR performances (STL, WSN, PFNR, and MTL) on the DAVIS50 dataset on the UVG8/17 dataset are lower than those on the UVG17 dataset. However, the performance trends of PFNR observed on the DAVIS50 dataset are consistent with the experimental results obtained from the UVG8/17 dataset. Regarding sequential and multi-task learning, the current performance falls short of achieving the target of 30 PSNR. This indicates a need for future work focused on designing model structures capable of sequence learning on large-scale datasets.

---

> ### Author Response · Authors · 2023-11-16
> **Compression Performances**
>
> - **Compression Performances**
>
>     To support the compression performances (Figure 2), we have prepared the sparsity-wise (c=10%, 30%, 50%, 70%) PSNR performances as shown in supplemental Table 17. Moreover, we have showed the progressive accumulated model capacity of PFNR which is less than 100\% (avg. 95\%), as shown in Fig. 3(b). Thus, the sparse and 8bit-quantized PFNR lead to parameter efficient PSNR performances. The core parts of the bit-wise quantization results are follows as.
>
>     | **PFNR, $f$-NeRV2** | **Sparsity** | **Avg. PSNR** | **Sparsity** | **Avg. PSNR** |
>     |---------------------|--------------|---------------|--------------|---------------|
>     | **bit=4**           | c=10.0 \%    | 24.79         | c=30.0 \%    | 24.19         |
>     | **bit=8**           | c=10.0 \%    | 28.09         | c=30.0 \%    | 29.33         |
>     | **bit=16**          | c=10.0 \%    | 28.10         | c=30.0 \%    | 29.36         |
>     | **bit=32**          | c=10.0 \%    | 28.10         | c=30.0 \%    | 29.36         |

---

> ### Author Response · Authors · 2023-11-16
> **Illustration of the proposed PFNR.**
>
> - **Illustration of the proposed methods**
>     As reviewer MyBt suggested, we have prepared the ablation studies on the proposed FSO to show its effectiveness for sequential neural implicit representations (please see our responses to MyBt's ablation requests).
>
>     - First, we show the performances of only real part (ignore an imaginary part) in f-NeRV2/3. The PSNR performances of only real part were lower than those of both real and imaginary parts in f-NeRV2/3. We infer that the imaginary part of the winning ticket improves the implicit neural representations.
>
>
>     - Second, we also investigate the effectiveness of only FSO without Conv.  in f-NeRV2/3. The PSNR performances were lower than FSO with Conv block. Therefore, the ensemble of FSO and Conv improves the implicit representations.
>
>
>     - Lastly, we investigate the effectiveness of sparse FSO in STL. The sparse FSO boots the PSNR performances in STL. These ablation studies further strengthen the effectiveness of FSO for sequential neural implicit representations.
>
> - **Visual Comparisions**
>
>     With limited space and capacity, we provided only important figures and attached more figures with public WSN code in the Appendix at submission. We will upload our revised script soon.
>
>     - Supplementary Figure 7 shows the video generation (from t=0 to t=2) with c = 30.0%, showing the PFNR’s output quality and PSNR scores on the UVG17 dataset. At the human recognition level, approximately 30 PSNR provides a little bit of blurred generated images as shown in WSN’s PSNR(29.26); if the PSNR score is greater than 30, we hardly distinguish the quality of sequential neural implicit representations (city session). Thus, our objective is to maintain the sequential neural implicit representation of the approximately 30 PSNR scores. We have achieved this target PSNR score on the UVG17 dataset.
>
>     - We also show the quantized and encoded PFNR's video generation results in Supplementary Figure 8. We demonstrate that a compressed sparse solution in FP8 (PFNR with $c=30.0 \%$, $f$-NeRV2) generates video representations sequentially without a significant quality drop. Additionally, the results of the FP4 showed that the compressed model can not create image pixels in detail.

---

> ### Author Response · Authors · 2023-11-16
> **Motivations and Improving scripts.**
>
> - **Question on Motivations**
>
>     - We agree with the reviewer that learning multiple unrelated videos at human and machine-level recognition is essential regardless of data relevancy or similarity.
>
>     - However, our algorithm assumed that even though the low similarity among unrelated videos, they share some representations in common at the neural implicit representation level as shown in Supplementary Figure 6 (Transfer Matrices); the upper triangular looks meaningless Forward Transfer (FWT), however, there might be some shared representations among uncorrelated videos and Figure 3(b) of reused weight (green, at least 10.0 \% for all video session) also support our claims (the remaining weights (90 \%) could be assigned to represent current session video).
>
>     - Therefore, we could conclude that PFNR leverages both aspects of relevancy and irrelevancy through adaptively reused weight selections for sequential video training.
>
>
> - **Minor Comments**
>     - Variant of PFNR
>
>         A straightforward variant of PFNR could depend on the layer position of FSO. In our experiments, the NeRV block layer of FSO determines its variants of PFNR, i.e., f-NeRV2 or f-NeRV3. Another variant of PFNR ignores imaginary parts, as shown in ablation studies. We have stated the importance of imaginary parts in FSO through the ablation studies despite its advantages of reducing parameters for the imaginary parts.
>
>
>     - Figure 1 size and typos
>
>         - As reviewers suggested, we have increased the font and diagram sizes in Figure 1.
>
>         - We have corrected some of the references. We would inspect any typos and references again for the final scripts.

---

> ### Comment · Reviewer_H5Ez · 2023-11-20
>
> Thank you for the detailed responses. I think they address most of my concerns. I'd like to update my overall ratings from 5 to 6.
>
> ~~Besides, I mentioned about more visual comparison in the reviews and if they can be further provided I think it would be help a lot to improve the quality of this work.~~I've seen some new figures in the revised pdf. Thank you for the update.

---

> > ### Author Response · Authors · 2023-11-21
> >
> > Thank you for your valuable feedback and insightful comments. Based on your suggestions, we would enhance our script's clarity and transparency and emphasize our work's novel aspects in the final version.

---

### Official Review · Reviewer_MyBt · 2023-11-01

**Soundness:** 3 good
**Presentation:** 3 good
**Contribution:** 3 good
**Rating:** 8
**Confidence:** 4

**Summary:**

The response provides a comprehensive analysis of a work focused on Neural Implicit Representation (NIR) for video data encoding. The work introduces a novel method, Progressive Fourier Neural Representation (PFNR), to improve the accumulation and transfer of neural implicit representations for complex video data across sequential encoding sessions. PFNR leverages a sparsified neural encoding in Fourier space, enabling better adaptation for future videos and lossless decoding for previous representations. The method shows impressive performance gains over continual learning baselines on UVG8/17 video sequence benchmarks.

Strengths of the work include the novel and straightforward concept of combining Fourier representation with sparsification and the method's superior performance over baselines with the same capacity. However, weaknesses are noted in the manuscript’s clarity, particularly in interpreting tables and comparing performance to baselines, as well as in the diverse configuration for Fourier transform.

Several questions are raised, seeking clarification on the commonality of NIR usage, details on Fourier transform configurations, baseline performances in Single Task Learning, comparisons to WSN, the meaning of outperforming an upper bound, and a typo in Table 3. These questions aim to probe deeper into the work’s methodology, performance, and presentation for a clearer understanding and evaluation.

**Strengths:**

1. The proposed idea, combining Fourier representation and its sparsification is a straightforward and novel concept.
2. With the same capacity, the proposed method provides better performance over baselines.

**Weaknesses:**

1. The target task seems to be small compared to the generality of the method.
2. The tables are a little hard to interpret.
3. The current version of the manuscript is missing comparable performance to its baselines.
4. The diverse configuration for Fourier transform.

**Questions:**

1. Is it common to use neural implicit representation (NIR)? As I know, many works adopt the term implicit neural representation (INR) rather than NIR.
2. The configuration for Fourier transform?
    - What is the temporal length of frames ($d_\nu$ in the paper right?) Does the $d_\nu$ affect the final performance?
    - Some sparsification protocols for Fourier transform ignore an imaginary part in both Fourier and inverse Fourier transform.
3. The performances of baselines in Single Task Learning (STL).
    - What is the performance of the proposed method with Single Task Learning (STL)? Does the proposed method impact negatively due to the proposed components for continual learning?
    - How about the performance only FSO without Conv block in NeRV block?
4. Comparison to WSN
    - What is the averaged PSNR and MS-SSIM performance of the proposed method and WSN on UVG8/17 with varying capacity? It is better to visualize in plot of the performance and capacity rather than table because the capacity of the model seems to be important for this setting. I conjecture that table 8 is the good starting point.
5. Meaning of upper bound
    - The proposed method, PFNR, outperforms the upper bound, MLT. What is the reason behind?
6. Typo
    - Table 3 shows the result of MS-SSIM while the caption has PSNR.

---

> ### Author Response · Authors · 2023-11-16
> **Ablation Studies on PFNR.**
>
> Thank you sincerely for your constructive feedback. We have summarized the reviewer's comments on the following main points: large-scale dataset, the performance of STL, ablation study on $f$-NeRV, the definition $d_v$ and its performances, and minor correction. Regarding these points, we provide detailed responses as follows. We have included all these discussions in our scripts to strengthen our novelty further and to be used broadly for various image enhancement and generation fields.
>
> - **Large-scale dataset (DAVIS50)**
>
>     Please refer to our responses to Reviewer H5Ez's request for large-scale experimental results.
>
> - **Performances of baselines in STL**
>
>     Our f-NeRV blocks positively impact single-task learning (STL) as follows. Mainly, f-NeRV3 boosts the PSNR performances of NeRV superiorly. This result could impact high-quality image-generation tasks.
>
>     | **UVG8**             | **1** | **2** | **3** | **4** | **5** | **6** | **7** | **8** | **avg. PSNR** |
>     |----------------------|-------|-------|-------|-------|-------|-------|-------|-------|---------------|
>     | STL, NeRV            | 39.66 | 36.28 | 38.14 | 42.03 | 36.58 | 29.22 | 37.27 | 31.45 | 36.33         |
>     | STL, NeRV, $f$-NeRV2 | 39.73 | 36.30 | 38.29 | 42.03 | 36.64 | 29.25 | 37.35 | 31.65 | 36.40         |
>     | STL, NeRV, $f$-NeRV3 | **42.75** | **37.65**  | **42.05** | **42.36**  | **40.01**  | **34.21**  | **40.15** |   **36.15**    |  **39.41**              |
>
>
> - **PFNR performance without Conv. in $f$-NeRV Blocks**
>
>     We show the performances of only FSO without Conv. block in NeRV block. The PSNR performances were lower than FSO with Conv block. Therefore, the ensemble of FSO and Conv improves the implicit representations. We have included this result in our final supplementary script.
>
>     | **UVG8**                               | **1**     | **2**     | **3**     | **4**     | **5**     | **6**     | **7**     | **8**     | **avg. PSNR** |
>     |-------------|-----------|-----------|-----------|-----------|-----------|-----------|-----------|-----------|---------------|
>     | **PFNR, c=50 \%, $f$-NeRV2** | **34.46** | **33.91** | **32.17** | **36.43** | **25.26** | **20.74** | **30.18** | **25.45** | **29.82**   |
>     | **PFNR, c=50 \%, $f$-NeRV2 w/o Conv.**   |   30.05   |  32.10    | 30.12     |	31.82  |	24.00  |	19.60  |	28.21  | 	24.47    |	27.54    |
>     | **PFNR, c=50 \%, $f$-NeRV2** | **36.45** | **35.15** | **35.10** | **38.57** | **28.07** | **23.06** | **32.83** | **27.70** | **32.12**   |
>     | **PFNR, c=50 \%, $f$-NeRV2 w/o Conv.**   |   35.46   | 35.06     |  34.98    |  38.23    | 28.00     | 22.98     | 32.57     | 27.45       | 31.84     |
>
>
> - **$f$-NeRV's performances without imaginary parts**
>
>     We show the performances of only real part (ignore an imaginary part) in f-NeRV2. The PSNR performances of only real part were lower than those of both real and imaginary parts, as shown in above. We infer that the imaginary part of the winning ticket improves the implicit neural representations. We have included this result in our final supplementary script.
>
>     | **UVG8**                                    | **1**     | **2**     | **3**     | **4**     | **5**     | **6**     | **7**     | **8**     | **Avg. PSNR** |
>     |---------------------------------------------|-----------|-----------|-----------|-----------|-----------|-----------|-----------|-----------|---------------|
>     | **PFNR, c=50 \%, $f$-NeRV2**                | **34.46** | **33.91** | **32.17** | **36.43** | **25.26** | **20.74** | **30.18** | **25.45** | **29.82**     |
>     | **PFNR, c=50 \%, $f$-NeRV2 w/o imag.** | 34.34     | 33.79     | 32.04     | 36.4      | 25.11     | 20.59     | 30.17     | 25.27     | 29.71         |
>     | **PFNR, c=50 \%, $f$-NeRV2**                | **36.45** | **35.15** | **35.10** | **38.57** | **28.07** | **23.06** | **32.83** | **27.70** | **32.12**     |
>     | **PFNR, c=50 \%, $f$-NeRV2 w/o imag.** | 35.66     | 34.65     | 34.09     | 37.95     | 25.80     | 21.94     | 32.17     | 26.91     | 31.15         |

---

> ### Author Response · Authors · 2023-11-16
> **Upper-bound (MTL) and Minor correction.**
>
> - **Meaning of upper-bound (MTL)**
>
>     We observed the two points regarding the results on our PFNR's comparable performances of MTL.
>
>     First, as the number of tasks increases, the performances of MTL tend to decrease (see, UVG8, UVG17, and DAVIS50). This is because uncorrelated individual samples could hinder the update gradients of MTL task models' parameters. From this, we speculate that the video sample's statistics (correlations, mean, and variance) should be considered to acquire the best performances of MTL in training.
>
>     Second, we have discussed the main representational difference between WSN and PFNR in Figure 4. The PFNR tends to capture local textures broadly (representations with high frequency and low variances) at lower layer (NeRV3) while while WSN focuses on local objects. This behavior of PFNR could leads to higher PSNR performances. To support our observations, we would conduct an additional representational analysis in terms of the frequency and variance in our future works.
>
> - **Minor points**
>
>     Thank you for your concerns about writing configuration, terminology, and typos. We would revise all configuration, terminology, and typos if possible.
>
>     - Writing configuration
>
>         Thank you for your constructive suggestions. Based on your feedback, we would move plots and images to supplementary and increase the portion of interpretable tables in the main script.
>
>     - Terminology (NIR or INR)
>
>         As the reviewer stated, some previous works follow the term, implicit neural representation (INR), but currently, a similar term, neural implicit representation (NIR) is also observed in various tutorials and works.
>
>     - Typos
>
>         We have revised the typo: Avg. MS-SSIM → Avg. PSNR in Table 3.

---

> ### Author Response · Authors · 2023-11-16
> **Temporal length and Averaged performances**
>
> - **Temporal length and Performances**
>
>     - the temporal length of frames ($d_v$ in Li et al., 2020a)
>
>         There is one difference between physical modeling and neural implicit representation settings. Following the previous definition (Li et al., 2020a), $d_v$ is the temporal length. In the NIR setting, the function $f$ learns a time-specific continuous output (an image) given a discrete-time index (a time).
>
>     - The final performance of $d_v$
>
>         If we would investigate the performances of $d_v$, we should consider the FPS of the video. As the reviewer (inAP) suggested, we investigated the performances of video samples according to their FPS. From our observations, the higher the FPS is, the better the PSNR is. Thus, our PFNR is a more time-specific continuous method than the others in physical modeling.
>
>
> - **Averaged PSNR Performances and Visualization**
>
>     The following tables show the averaged PSNR and MS-SSIM performances of WSN on UVG8/17 with varying capacity (c=10%, 30%, 50%, 70%). We have included more generated images in the Supplementary.
>
>     | UVG8            | Avg. PSNR | Avg. MS-SSIM |
>     |-----------------|-----------|--------------|
>     | STL, NeRV       | 36.33     | 0.97         |
>     | EWC             | 12.58     | 0.30         |
>     | iCaRL           | 26.51     | 0.80         |
>     | ESMER           | 21.92     | 0.61         |
>     | WSN             | 27.26     | 0.84         |
>     | PFNR, $f$-NeRV2 | 29.82     | 0.90         |
>     | PFNR, $f$-NeRV3 | 32.12     | 0.93         |
>     | MTL             | 30.78     | 0.91         |
>
>
>     | UVG17           | Avg. PSNR | Avg. MS-SSIM |
>     |-----------------|-----------|--------------|
>     | STL, NeRV       | 36.94     | 0.97         |
>     | EWC             | 11.38     | 0.30         |
>     | iCaRL           | 26.67     | 0.65         |
>     | ESMER           | 20.95     | 0.62         |
>     | WSN             | 26.42     | 0.82         |
>     | PFNR, $f$-NeRV2 | 29.36     | 0.90         |
>     | PFNR, $f$-NeRV3 | 31.72     | 0.94         |
>     | MTL             | 29.42     | 0.90         |

---

> > ### Comment · Reviewer_MyBt · 2023-11-20
> >
> > Thank you for your detailed explanations and extensive experiments.
> > I increased my score from 6 to 8 because the rebuttal solved most of my concerns.
> > I hope the comments from all reviewers will be included in the future version of the paper.

---

> > > ### Author Response · Authors · 2023-11-21
> > >
> > > Thank you for your valuable feedback and comments. We would improve the entire script to be more transparent and strengthen our novel points in the final version, further.

---

### Official Review · Reviewer_uKbd · 2023-11-02

**Soundness:** 3 good
**Presentation:** 2 fair
**Contribution:** 3 good
**Rating:** 6
**Confidence:** 4

**Summary:**

This paper investigates the neural implicit representation in continual learning settings and proposes a novel method to encode the learned knowledge compactly. With this design, the model becomes able to accumulate neural representations for multiple videos with few decoding losses for previous videos. Experiments on UVG8/17 verifies its effectiveness.

**Strengths:**

1. The proposed learning scenario is very practical in real applications, that is, encoding multiple videos continually via a neural implicit representation. With this setting, the generalization ability of NIR can be evaluated.

2. I like the idea that incorporates the Lottery Ticket Hypothesis to find subnetworks for each video session. Frozing these subnetworks proves to be an effective way to encode existing knowledge about previous videos.

3. The designed experiments verify its ability to adapt to new training sessions. With the proposed methods, the performance achieved significant improvements.

**Weaknesses:**

1. The writing is not so clear. It's hard to learn the connection of Sec 3.1 with other sections.

2. The experiments miss one important point. All tables show the results of the newly added sessions. However, to verify the "lossless decoding" stated in the abstract, previous videos need also to be evaluated. Otherwise, the reported metrics only verify the quick adaptation ability of the proposed method.

3. There are some neglected details.
- At the 2nd line of Sec 3, the cited paper is E-NeRV instead of NeRV.
- In the caption of Fig 1, the symbol $H_N$ is used without definition and no further usage.
- At the line before Eq 3, should it be "session s" instead of "session t"?
- I suggest moving Algorithm 1 to the same page as Sec 3.2, where it is referred.

**Questions:**

In Algorithm 1, the operation at line 7 is not differentiable, so how to calculate $\partial L / \partial \rho$?
I'm willing to change my score if all my issues can be solved

---

> ### Author Response · Authors · 2023-11-16
> **Lossless decoding (forget-free)**
>
> Thank you sincerely for your constructive feedback. We have summarized the reviewer's comments on three main points: lossless decoding (forget-free), Straight-through estimator, and improving writing. Regarding the three points, we provide detailed responses as follows.
>
> - **Lossless decoding (forget-free)**
>
>     We have prepared the PFNR transfer matrices on the UVG17 dataset to show the lossless decoding, as shown in Supplementary Figure 6. The lower triangular estimated by each session subnetwork denotes that our PFNR is a forget-free method, and the upper triangular calculated by the current session subnetwork denotes the video similarity between the source and the target videos. The following PFNR transfer matrice results from PFNR, c=30.0\%, $f$-NeRV3.
>
>     |     | S1    |       |       |       | S5    |       |       |       |       | S10   |       |       |       |       | S15   |       |       |
>     |-----|-------|-------|-------|-------|-------|-------|-------|-------|-------|-------|-------|-------|-------|-------|-------|-------|-------|
>     | S1  | **33.63** | 7.57  | 8.75  | 9.97  | 8.70  | 7.28  | 9.77  | 9.49  | 9.72  | 8.88  | 8.48  | 10.03 | 9.98  | 9.29  | 7.36  | 7.50  | 7.83  |
>     |     | **33.63** | **39.24** | 12.44 | 9.45  | 5.90  | 7.02  | 9.60  | 9.63  | 8.91  | 11.42 | 7.29  | 9.59  | 9.84  | 7.30  | 5.27  | 4.48  | 10.41 |
>     |     | **33.63** | **39.24** | **34.21** | 11.08 | 5.92  | 6.91  | 10.49 | 11.13 | 9.91  | 14.91 | 7.36  | 11.29 | 11.54 | 7.48  | 5.28  | 4.73  | 12.21 |
>     |     | **33.63** | **39.24** | **34.21** | **37.79** | 7.96  | 8.59  | 10.53 | 11.55 | 10.69 | 11.19 | 9.23  | 10.49 | 11.58 | 8.78  | 7.29  | 6.60  | 9.48  |
>     | S5  | **33.63** | **39.24** | **34.21** | **37.79** | **34.05** | 8.41  | 8.33  | 6.70  | 8.56  | 5.58  | 9.30  | 7.08  | 7.02  | 10.75 | 12.31 | 10.85 | 5.30  |
>     |     | **33.63** | **39.24** | **34.21** | **37.79** | **34.05** | **27.17** | 8.08  | 7.69  | 9.18  | 6.90  | 9.56  | 6.97  | 7.35  | 8.83  | 7.53  | 6.07  | 6.99  |
>     |     | **33.63** | **39.24** | **34.21** | **37.79** | **34.05** | **27.17** | **38.17** | 10.02 | 11.21 | 11.00 | 8.74  | 11.52 | 11.55 | 8.86  | 7.36  | 5.93  | 10.54 |
>     |     | **33.63** | **39.24** | **34.21** | **37.79** | **34.05** | **27.17** | **38.17** | **29.79** | 10.43 | 11.30 | 8.10  | 10.95 | 11.73 | 7.78  | 6.05  | 5.37  | 10.19 |
>     |     | **33.63** | **39.24** | **34.21** | **37.79** | **34.05** | **27.17** | **38.17** | **29.79** | **26.56** | 10.55 | 9.56  | 11.18 | 11.58 | 9.48  | 7.61  | 6.16  | 10.24 |
>     | S10 | **33.63** | **39.24** | **34.21** | **37.79** | **34.05** | **27.17** | **38.17** | **29.79** | **26.56** | **36.18** | 7.44  | 11.61 | 12.07 | 7.37  | 4.97  | 4.27  | 14.71 |
>     |     | **33.63** | **39.24** | **34.21** | **37.79** | **34.05** | **27.17** | **38.17** | **29.79** | **26.56** | **36.18** | **22.97** | 7.50  | 7.63  | 9.08  | 8.74  | 7.21  | 7.10  |
>     |     | **33.63** | **39.24** | **34.21** | **37.79** | **34.05** | **27.17** | **38.17** | **29.79** | **26.56** | **36.18** | **22.97** | **24.36** | 14.14 | 7.83  | 6.31  | 5.30  | 11.44 |
>     |     | **33.63** | **39.24** | **34.21** | **37.79** | **34.05** | **27.17** | **38.17** | **29.79** | **26.56** | **36.18**| **22.97** | **24.36** | **32.50** | 8.09  | 6.26  | 5.33  | 11.42 |
>     |     | **33.63** | **39.24** | **34.21** | **37.79** | **34.05** | **27.17** | **38.17** | **29.79** | **26.56** | **36.18**| **22.97** | **24.36** | **32.50** | **30.22** |  8.39  | 8.36  | 6.93  |
>     | S15 | **33.63** | **39.24** | **34.21** | **37.79** | **34.05** | **27.17** | **38.17** | **29.79** | **26.56** | **36.18**| **22.97** | **24.36** | **32.50** | **30.22** | **27.62** | 9.32  | 4.75  |
>     |     | **33.63** | **39.24** | **34.21** | **37.79** | **34.05** | **27.1** | **38.17** | **29.79** | **26.56** | **36.18** | **22.97** | **24.36** | **32.50** | **30.22** | **27.62** | **29.15** | 3.64  |
>     |     | **33.63** | **39.24** | **34.21** | **37.79** | **34.05** | **27.17** | **38.17** | **29.79** | **26.56** | **36.18** | **22.97** | **24.36** | **32.50** | **30.22** | **27.62** | **29.15** | **35.68** |

---

> ### Author Response · Authors · 2023-11-16
> **Straight-through estimator to update weight schore $\rho^{\ast}$ and Improving scripts.**
>
> - **Straight-through estimator for $\frac{\partial L}{\partial \rho^{\ast}}$**
>
>     To update the weight score $\rho^{\ast}$, we have used the Straight-through estimator [1,2,3], as stated in Sec. 3.2. The Straight-through estimator estimates the gradients of a function. Specifically, it ignores the derivative of the threshold function (binary mask) and passes on the incoming gradient as if the function were an identity function. So, the threshold function (binary mask) is bypassed in the backward pass.
>
>     [1] Neural networks for machine learning, Hinton, 2012
>
>     [2] Estimating or propagating gradients through stochastic neurons for conditional computation- CoRR2013
>
>     [3] What’s hidden in a randomly weighted neural network?- CVPR2020
>
> - **Improving writing**
>
>     Based on the reviewer's constructive feedback, we have revised the descriptions (particularly, Sec. 3.1.). The minor revised points are as follows:
>
>     - We have revised the typos: the cited paper (NeRV), "session t".
>     - We have redefined the $H_N$ and used to explain the trainable function $f$. For simplicity equation, we ignore the $H_N$.
>     - We have moved Algorithm 1 to the same page as Sec. 3.2.
>
>     We would improve the entire script to be more transparent and strengthen our novel points in the final scripts. We will upload our revision soon.

---

> > ### Comment · Reviewer_uKbd · 2023-11-23
> > **Issues well solved**
> >
> > All my issues have been solved, so I'm willing to raise my score

---

### Official Review · Reviewer_inAP · 2023-11-02

**Soundness:** 3 good
**Presentation:** 2 fair
**Contribution:** 3 good
**Rating:** 8
**Confidence:** 4

**Summary:**

This article addresses the existing problem in Neural Implicit Representation (NIR) that these models specialize in learning only one mapping between target data and fail to generalize when implemented over more data, thus limiting scalability. The authors propose a modification on top of the NeRV pipeline [Chen et al, 2021a]. In order to capture information while doing a fine discretization of model parameters, the authors transform the weights to the Fourier space, on the lines of the work done by [Li et al., 2020a]. Experimentation has been done on Video Task-incremental Learning and comparisons have been made against relevant baselines. The experiments closely follow NeRV [Chen et al, 2021a] and HNeRV [Chen et al., 2023] to make fair comparisons. Results are satisfactory and visualizations look good.


Hao Chen, Bo He, Hanyu Wang, Yixuan Ren, Ser Nam Lim, and Abhinav Shrivastava. Nerv: Neural representations for videos. Advances in Neural Information Processing Systems, 34:21557–21568, 2021a.

Zongyi Li, Nikola Kovachki, Kamyar Azizzadenesheli, Burigede Liu, Kaushik Bhattacharya, Andrew Stuart, and Anima Anandkumar. Fourier neural operator for parametric partial differential equations. arXiv preprint arXiv:2010.08895, 2020a

Hao Chen, Matt Gwilliam, Ser-Nam Lim, and Abhinav Shrivastava. Hnerv: A hybrid neural representation for videos. arXiv preprint arXiv:2304.02633, 2023.

**Strengths:**

The article benefits from generally good writing, appropriate use of mathematical language.

Background literature survey is highly appropriate and extensive.

The article builds on top of a NeRV pipeline, addressing relevant and interesting shortcomings. The rationale behind the use of Fourier Subneural Operators has been developed clearly.

Extensive experimentation and provision of visual results in the paper and the supplementary file also adds to the strengths of the paper. Results are satisfactory.

The ability of the model to generate videos has also been shown in the supplementary document which only adds to the broader impacts of the research done by the authors.

**Weaknesses:**

The methodology section may appear somewhat difficult to some readers without the necessary background in the domain. This however is only a minor weakness that does not affect the final rating.

It is not very clear if the authors used the exact frame sizes of the videos as available, or if the authors did any spatial or temporal downsampling on the videos to fit the model training pipeline, as is usual with most papers pertaining to video data.

It appears that the model may be limited by computational expenses as the datasets used are that of short video clips.

**Questions:**

How does the proposed architecture compare with existing baselines in terms of computational requirement? Does the FSO module increase the computational requirements significantly from the baseline NeRV?

How many frames for each video was considered during training? Can the authors elaborate the video frame resolution vs frame rate trade-off i.e. what’s the maximum number of frames that can be considered at a time at a certain resolution or what’s the maximum resolution that can be processed at a certain number of frames?

---

> ### Author Response · Authors · 2023-11-16
> **Dataset statistics and Performances.**
>
> Thank you sincerely for your constructive feedback. We have summarized the reviewer's comments on two main points: data statistics of FPS and PSNR and computational expenses. Regarding the two points, we provide detailed responses as follows.
>
> - Dataset statistics
>
>     We have prepared the dataset statistics of the UVG Videos as shown in supplementary tables 5 and 6. For example, the UVG17 Video Sessions are below. We provide the frame size and resolutions for training and inferences.
>
> - FPS and PSNR performances.
>
>     As shown in supplementary tables 5 and 6, we have extracted sample frames according to each video’s FPS and Length (sec) during training. There are two kinds of videos: 300 and 600 frames. We train each video for 150 epochs. Statistically, the average PSNR of 120 FPS was better than those of 50 FPS, as shown below. When considering 30 PSNR is known as a sufficient resolution, 50 FPS-based sequence training could be adequate using the proposed PFNR. Moreover, the number of frames does not seem to be a critical factor to PSNR when considering that the PSNR of video session 1 (bunny), with 132 (720 x 1080) frames, is greater than the 30 PSNR score.  We would include this results to our scripts.
>
>     | **UVG17**  | **FPS / Resolution** | **Avg. PSNR** | **FPS / Resolution** | **Avg. PSNR** |
>     |------------|----------------------|---------------|----------------------|---------------|
>     | **STL**    | 50 / 1920 x 1080     | **35.97**     | 50 / 1920 x 1080     | 37.56         |
>     | **iCaRL**  | 50 / 1920 x 1080     | **21.94**     | 50 / 1920 x 1080     | 21.06         |
>     | **ESMER**  | 120 / 1920 x 1080    | **21.43**     | 50 / 1920 x 1080     | 19.25         |
>     | **WSN**    | 120 / 1920 x 1080    | **27.01**      | 50 / 1920 x 1080     | 25.95         |
>     | **PFNR, $f$-NeRV2** | 120 / 1920 x 1080 | **29.65**  | 50 / 1920 x 1080     | 28.36         |
>     | **PFNR, $f$-NeRV3** | 120 / 1920 x 1080 | **31.53**  | 50 / 1920 x 1080     | 31.47         |
>     | **MTL**           | 120 / 1920 x 1080 | **29.64**  | 50 / 1920 x 1080     | 28.83         |

---

> ### Author Response · Authors · 2023-11-16
> **Computational expanses and improvement of scripts.**
>
> - Computational expanses
>
>     We train and test two baselines (NeRV, ESMER) with $f$-NeRV2 using one GPU (TITAN V, 12G) with a single batch size to investigate the computational expenses and decoding FPS on the UVG8 dataset, as shown below. In STL, NeRV with $f$-NeRV2 costs more computational times than NeRV. In VCL, memory buffer-based methods, i.e., ESMER, cost more training time since they replay the memory buffer in sequential training. On the other hand, architecture-based methods, i.e., PFNR, provide parameter-efficient, faster, forget-free solutions in training while cost computation expenses in the decoding process. Considering these limitations and advantages, we would find a more parameter-efficient FSO algorithm in future work.
>
>     | UVG8                 | Training time [hours] | Decoding FPS |
>     |----------------------|-----------------------|--------------|
>     | STL, NeRV            | 13.10 [h]             | 56.88        |
>     | STL, NeRV, $f$-NeRV2 | 25.66 [h]             | 31.72        |
>     | ESMER                | 101.71 [h]            | 56.93        |
>     | PFNR, $f$-NeRV2      | 34.52 [h]             | 31.55        |
>
>
> - Improvmentment of domain backgrounds.
>
>     We would improve domain backgrounds (particularly in Sec. 3.1. FSO) for readers in various research fields. The reviewer's constructive feedback and comments have strengthened our novelty and guided future work direction. We would include all these discussions in our scripts.

---

> > ### Comment · Reviewer_inAP · 2023-11-22
> >
> > I am comfortable to keep my current score.

---

### Author Response · Authors · 2023-11-19
**Responses to the reviewers' comments.**

Dear All Reviewers,

We sincerely appreciate your constructive feedback and have addressed each of your main points as follows:

For reviewer inAP,
 - Detailed dataset statistics for UVG Videos are now in supplementary tables 5 and 6.

 - We have discussed the relationship between FPS and PSNR performances.

 - Further explanation regarding computational expenses has been added.

For reviewer MyBt,
 - Extensive experiments on the DAVIS50 dataset have been conducted.

 - The effectiveness of the proposed PFNR (Progressive Fourier Neural Representation) is demonstrated through comprehensive ablation studies.

 - We have included the averaged performances in our analysis.

 - Responses to questions about the Upper-bound and temporal length have been prepared.

For reviewer uKbd,
 - PFNR transfer matrices on the UVG17 dataset have been prepared to prove lossless decoding.

 - We have added the explaination of Straight-through estimator for $\frac{\partial L}{\partial \rho^{\ast}}$.

For reviewer H5Ez,
 - We have prepared additional results on a large-scale sequential video dataset, the Densely Annotation Video Segmentation (DAVIS50).

 - Core aspects of quantization performances have been added to our discussion.

 - More generated images are now visible in the Supplementary materials.

 - We have elaborated on the motivation behind our research.

 - Corrections to figure sizes and script revisions have been made.

We have carefully revised our manuscript in light of the insightful feedback provided by the reviewers. We believe that our responses and the additional results comprehensively address the concerns raised. These revisions will significantly strengthen the paper.

---

### Meta-Review · Area_Chair_KhFJ · 2023-12-09

**Metareview:**

This paper proposed Progressive Fourier Neural Representation (PFNR) method improves upon NIR by creating adaptive, compact modules in Fourier space for video encoding, allowing for better adaptation for future videos and preserving previous representations, showing impressive performance gains in benchmarks. The reviewers find the paper is solving a practical problem, and the experiments are extensive and all recommend accepting the paper. The AC agrees with the reviewer on recommend acceptance.

**Justification For Why Not Higher Score:**

The paper presents solid results, but reviewers are generally concerned that the presentation of the work is not that easy to understand.

**Justification For Why Not Lower Score:**

All reviewers agree to accept the paper.

---

### Decision · Program_Chairs · 2024-01-16

Accept (poster)